# SWT-Bench: Testing and Validating Real-World Bug-Fixes with Code Agents

**Niels Mündler[1], Mark Niklas Müller[1,2], Jingxuan He[1], Martin Vechev[1]**

[1] Department of Computer Science, ETH Zurich      [2] LogicStar.ai

{niels.muendler, mark.mueller, jingxuan.he, martin.vechev}@inf.ethz.ch [1]
mark@logicstar.ai [2]

## Abstract

Rigorous software testing is crucial for developing and maintaining high-quality code, making automated test generation a promising avenue for both improving software quality and boosting the effectiveness of code generation methods. However, while code generation with Large Language Models (LLMs) is an extraordinarily active research area, test generation remains relatively unexplored. We address this gap and investigate the capability of LLM-based Code Agents to formalize user issues into test cases. To this end, we propose a novel benchmark based on popular GitHub repositories, containing real-world issues, ground-truth bug-fixes, and golden tests. We find that LLMs generally perform surprisingly well at generating relevant test cases, with Code Agents designed for code repair exceeding the performance of systems designed specifically for test generation. Further, as test generation is a similar but more structured task than code generation, it allows for a more fine-grained analysis using issue reproduction rate and coverage changes, providing a dual metric for analyzing systems designed for code repair. Finally, we find that generated tests are an effective filter for proposed code fixes, doubling the precision of SWE-AGENT. We release all data and code at github.com/logic-star-ai/SWT-Bench.

## 1 Introduction

As the complexity of software systems increases, rigorous testing is becoming more important than ever to ensure their reliability and correctness. However, while a large portion of these tests aims to reproduce previously reported *issues* (Kang et al., 2023), such issue reproduction is often disliked by professional developers (Straubinger & Fraser, 2023). Therefore, automatic generation of tests reproducing such issues from informal natural language descriptions is a promising path toward improving both code quality and developer productivity. Finally, generated tests can be leveraged as formal specifications to boost the effectiveness of automatic code repair tools (Chen et al., 2023a).

However, while automatic code generation, in particular using Code Agents, is an extremely active research area (i.e. Yang et al. (2024); Tao et al. (2024); Zhang et al. (2024); Bouzenia et al. (2024b); OpenDevin (2024); Bouzenia et al. (2024a); Schäfer et al. (2024); Alshahwan et al. (2024a)), there is comparatively little work investigating automatic test generation directly. Indeed, while prior work has proposed methods based on symbolic execution (Lukasczyk & Fraser, 2022), specialized transformers (Tufano et al., 2020), and general-purpose LLMs (Li et al., 2023; Alshahwan et al., 2024b; Kang et al., 2023, 2024; Chen et al., 2023b), Code Agents have not been considered in this context, and even less work is applicable to the issue reproduction setting. Finally, large-scale, diverse test-generation datasets are lacking for Python, which is one of the most popular programming languages at the time of writing (TIOBE, 2024; PYPL, 2024) and a focus of Code Agent research.

38th Conference on Neural Information Processing Systems (NeurIPS 2024).

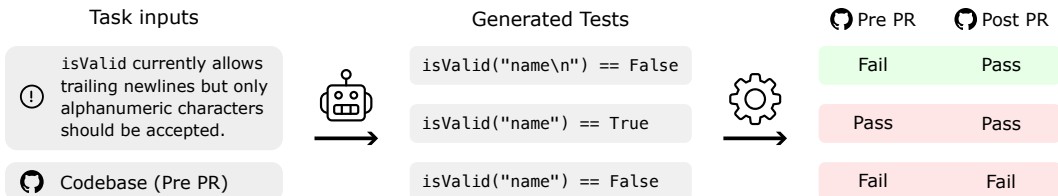

Figure 1: Evaluation of an SWT-BENCH instance. Given an issue description in natural language and the corresponding codebase, the task is to generate tests that reproduce the issue. We considered a test to reproduce the issue if it fails on the codebase before the pull request (PR) is accepted, i.e., before the golden patch is applied, but passes after. We call this a fail-to-pass test ($F \rightarrow P$).

**A Benchmark for Test Generation** In this work, we propose SWT-BENCH, a novel and comprehensive dataset for test generation with the goal of issue reproduction in Python. SWT-BENCH contains over 1 900 samples, each consisting of a GitHub issue, a golden code patch resolving the issue by adjusting the code, and a set of golden reference tests, obtained by transforming the popular SWE-BENCH (Jimenez et al., 2023) from code repair to test generation. We leverage the fact that any code repair task can be transformed into a test generation task, even in the absence of golden tests, by utilizing the golden code patch for evaluation. Concretely, for every generated test, we determine whether it reproduces the described issue, by checking whether it fails on the original repository but passes after the golden patch is applied. The golden reference tests, used in SWE-BENCH for the evaluation of code repair performance, are solutions in this test generation setting. We illustrate this evaluation process of SWT-BENCH in Fig. 1. Further, we report the coverage of the code modified by the golden patch as a more fine-grained evaluation metric for generated tests.

**Benchmarking Test Generation Methods** We evaluate various existing test generation approaches on SWT-BENCH, including directly prompting state-of-the-art LLMs to generate tests for the given issue, a state-of-the-art issue reproduction method LIBRO (Kang et al., 2023), and different Code Agents adapted to the task of test generation (Yang et al., 2024; Zhang et al., 2024; Aider, 2024). Interestingly, we find that despite being designed for code repair, the Code Agent SWE-AGENT outperforms non-agent methods at test generation, both reproducing more issues and achieving higher coverage, and generally find all agents to perform strongly in both areas. However, we still observe significant complementarity between the different approaches, with an ideal ensemble of the best four methods solving 71% more samples than the best single method. Further, while the performance on code repair and test generation is generally correlated, this does not hold on a per-sample basis. This indicates that reproducing an issue with a test and fixing this issue are distinct tasks of different difficulty. Finally, we find that generated tests can serve as a strong signal for the correctness of proposed code fixes, with SWE-AGENT achieving over twice the precision on fixes that pass self-generated tests that failed before the fix was applied.

**Key Contributions** Our key contributions are:

- We introduce SWT-BENCH, a new benchmark for test-based issue reproduction based on an extensive dataset of real-world software repositories, user issues, code patches, and test cases (§3).
- We propose to adapt Code Agents to the task of test generation for issue reproduction (§4).
- We provide an extensive evaluation of SWT-BENCH, and demonstrate that, while issue reproduction is generally hard, Code Agents perform well, outperforming prior methods (§5).

## 2 Related Work

**Code Datasets** Over recent years, a variety of code datasets such as HumanEval (Chen et al., 2021), APPS (Hendrycks et al., 2021), and MBPP (Austin et al., 2021) have been proposed to assess the capabilities of code synthesis and repair systems (Lin et al., 2017; Li et al., 2022). However, these largely focus on interview-style coding challenges or function-level code synthesis and do not capture the complexity of real-world codebases. Further, they have been shown to often include insufficient test cases to properly assess the correctness of the generated code (Liu et al., 2023a).

Recently, a range of repository-level code-generation benchmarks (Liu et al., 2023b; Jain et al., 2024) including the popular SWE-BENCH (Jimenez et al., 2023) have emerged, as modern LLMs

began to saturate the simpler function-level benchmarks. However, none of these benchmarks were designed to assess test generation.

The only dataset for reproducing bugs based on real-world issues, Defects4J (Just et al., 2014), focuses on Java, is outdated, limited in size, and contains only short bug descriptions rather than detailed issue reports. In contrast, SWT-BENCH is based on Python, which is better supported by modern Code Agents, contains more recent issue reports, and is significantly larger.

**Automated Unit Test Generation**   Many approaches have been suggested to automate (unit) test generation leveraging symbolic execution (Lukasczyk & Fraser, 2022), specialized transformers (Tufano et al., 2020), and general purpose LLMs (Li et al., 2023; Alshahwan et al., 2024b; Kang et al., 2023; Tufano et al., 2020; Kang et al., 2024; Schäfer et al., 2024; Alshahwan et al., 2024a; Chen et al., 2023b). Depending on their focus, they can be used to increase test coverage (Alshahwan et al., 2024b; Schäfer et al., 2024), find edge cases (Lukasczyk & Fraser, 2022), or reproduce reported issues (Kang et al., 2023). Issue-reproducing tests are especially interesting, as they can be used to validate automatically generated code repair candidates and thus improve the precision of code repair systems (Chen et al., 2023a). However, most test-generation approaches are not applicable to issue reproduction. We therefore evaluate the most recent applicable method, LIBRO (Kang et al., 2023), and a range of other LLM-based baselines on SWT-BENCH.

**Code Agents**   Over the last year, LLMs have been equipped with tools to observe and interact with their environment over multiple turns and preserve a state across these turns (Wang et al., 2024). These so-called agents have proven successful on a range of complex tasks, including code repair and synthesis (Bouzenia et al., 2024b; OpenDevin, 2024; Zhang et al., 2024; Yang et al., 2024; Tao et al., 2024; Bouzenia et al., 2024a; Aider, 2024). Such Code Agents can typically search, read, and edit code using an agent computer interface (ACI) (Yang et al., 2024). In this work, we leverage such Code Agents for generating issue-reproducing tests by changing their instructions.

## 3   Benchmarking Test Generation

In this section, we outline the structure of the proposed benchmark, SWT-BENCH, and how we leverage it to measure the capabilities of LLMs and Code Agents for test generation.

### 3.1   Notation and Definitions

We first introduce the notation to describe codebases, their test suites, and changes to these codebases in the form of patches. We denote a codebase $R$ after applying patch $X$ as $R \circ X$. Several patches can be applied sequentially, i.e. $R \circ X \circ Y$ is the codebase $R$ after applying a first patch $X$ and then a second one $Y$. When a code patch $X$ is applied to $R$, a set of tests $T$ can be used to check the correctness of the applied patch.

A single test $s$ can either pass (P) or fail (F) after we execute it within the context of codebase $R$. We consider a test to fail if an error is thrown during its execution, e.g., an `AssertionError` or `ValueError`. Such test errors frequently occur if $R$ lacks or misimplements the functionality targeted by the test. They can also occur due to other reasons, such as incorrect syntax or formatting of the test $s$. Conversely, a test passes when running the test triggers no error. We define this process as an execution function: $\mathrm{exec}(s, R) \in \{P, F\}$.

We consider a test $s$ to reproduce a described issue $I$ of $R$, which is resolved by patch $X$ if it fails on the original codebase (i.e. $\mathrm{exec}(s, R) = F$) but passes on the patched codebase (i.e. $\mathrm{exec}(s, R \circ X) = P$). We denote these fail-to-pass tests with $F \to P$ and define $F \to F$, $P \to P$, and $P \to F$ tests similarly. If a test transitions from failing on $R$ to any state on $R \circ X$, we denote it as $F \to \times$ and vice versa for $\times \to F$. Further, we consider a set of tests $T$ to be successful at reproducing the issue $I$, if it contains at least one $F \to P$ test and no $\times \to F$ test, or equivalently $\exists s \in T, \mathrm{exec}(s, R) = F \wedge \forall s \in T, \mathrm{exec}(s, R \circ X) = P$.

### 3.2   Benchmark Overview

To construct SWT-BENCH, we leverage the same underlying data as SWE-BENCH (Jimenez et al., 2023) and summarize its three-stage construction process here for completeness.

1. Scrape a total of $\sim 90\,000$ pull requests (PRs) from 12 popular open-source Python repositories from GitHub.

2. Filter PRs to only include those that were merged, resolved a GitHub issue, and made changes to at least one test file.

3. Filter PRs to feature at least one $F \rightarrow P$ test, removing PRs that result in installation or runtime errors.

This results in $2\,294$ task instances, each consisting of a GitHub issue, a golden patch $X^*$ fixing the issue, and a set of golden reference tests $T^*$.

However, we find that for 311 instances, the golden patch can not be evaluated without errors or does not fix the described issue reliably, i.e., some tests of $T^*$ fail on $R \circ X^*$. The main reasons are flaky test suites, e.g., django cases where HTTP requests sometimes return 500 Internal Server Error although the related code was not changed, erroneous test suite setup, e.g., the test suite tool tox not allowing external tools invoked in the sphinx setup, and time outs, e.g., when slow tests in the sympy library are run. We exclude these, leaving a total of $1\,983$ instances in SWT-BENCH. To enable cheaper evaluation we create SWT-BENCH-LITE, a subset of 276 issues, corresponding to SWE-BENCH-LITE.

Table 1: Characterization of different attributes of SWT-BENCH instance.

|  |  | Mean | Max |
|---|---|---|---|
| Issue Text | # Words | 318.0 | 8756 |
| Codebase | # Files | 1563.0 | 2757 |
|  | # Lines | 337K | 772K |
| Existing Tests | # $F \rightarrow P$ | 0.0 | 4 |
|  | # $F \rightarrow F$ | 5.0 | 183 |
|  | # $P \rightarrow P$ | 116.7 | 4837 |
|  | # $P \rightarrow F$ | 1.5 | 607 |
|  | # Total | 123.2 | 4842 |
|  | Coverage | 70.1% | 100% |
| Golden Tests | # $F \rightarrow P$ | 2.0 | 958 |
|  | # $P \rightarrow P$ | 0.9 | 339 |
|  | # Added | 2.9 | 958 |
|  | # Removed | 0.3 | 104 |
|  | # Files Ed. | 1.5 | 15 |
|  | # Lines Ed. | 31.8 | 581 |

We summarize key statistics of SWT-BENCH in Table 1 and show its repository composition in Fig. 2. While issue descriptions are on average only 318 words long, the longest one reaches $8\,756$ words. Generally, repository complexity is high with on average over $1\,500$ files and over $300\,000$ lines of code. Many repositories feature large test suites of $> 120$ and up to $4\,800$ tests, already covering 70% of the lines in the to-be-patched code. Most of these existing tests are unaffected by the golden patch with basically no $F \rightarrow P$ and only 1.5 $P \rightarrow F$ tests on average. The golden tests remove on average 0.3 tests and add another 2.9 new test cases, of which roughly two-thirds are $F \rightarrow P$. The test patches edit on average 31.8 lines in 1-2 files. Due to the filtering for unresolved issues during dataset curation, no golden tests are $F \rightarrow F$ or $P \rightarrow F$.

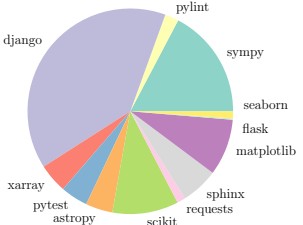

Figure 2: Distribution of SWT-BENCH instances over GitHub repositories.

### 3.3 Metrics

We propose two main metrics to evaluate the test generation performance of any method; Success rate ($\mathcal{S}$) and change coverage ($\mathcal{C}$), described below. We further introduce the necessary but insufficient property of patch well-formedness ($\mathcal{W}$).

**Success Rate** The success rate $\mathcal{S}$ measures the portion of instances where the generated tests $T$ reproduced the issue according to the definition in §3.1, i.e. at least one test in $T$ transitions from failing to passing and none fail after applying the patch. This is the most important performance measure, as the presence of $F \rightarrow P$ and the absence of $\times \rightarrow F$ tests are key for test-driven development and automatic code generation. We further report the portion of instances for which at least one Fail-to-Pass ($F \rightarrow P$), Fail-to-Any ($F \rightarrow \times$), and Pass-to-Pass ($P \rightarrow P$) was generated. While $F \rightarrow \times$ tests, i.e., all tests that fail on the original codebase, are not necessarily desirable, only $F \rightarrow \times$ tests can result in the reproducing $F \rightarrow P$ test, whereas $P \rightarrow \times$ tests can never reproduce an issue. As $F \rightarrow \times$ can further be identified without knowledge of the golden code patch, generation methods can aim to always produce an $F \rightarrow \times$ test. Finally, $P \rightarrow P$ tests indicate that the model generated well-formed and valid, but unrelated tests.

**Change Coverage**   Coverage is an important metric to determine what portion of a codebase is tested. While path coverage measures this optimally, the exponential number of paths makes it infeasible in practice. We thus follow common practice, and instead measure line coverage. As we aim to specifically test the code described in the issue text, we consider only the coverage of the changes made by the golden code patch. Further, we observe that patches may include portions of non-executable lines, e.g. documentation or configuration files, and exclude them. Specifically, we consider all lines that are executed by the original test suite $T^R$ or the golden test suite $T^*$ on both $R$ and $R \circ X^*$ to be executable, and track coverage of such executable lines.

Finally, we consider both the coverage of removed (including modified) lines of code in the original codebase and added (including modified) lines of code in the patched codebase, illustrated in Fig. 3.

Formally, given the number of times $\mathcal{C}^R_{T^R}(l) \in \mathbb{Z}^{\geq 0}$ a specific line of code $l$ was executed when running the test suite $T^R$ on codebase $R$, we define the executable lines of the patch $X$ as

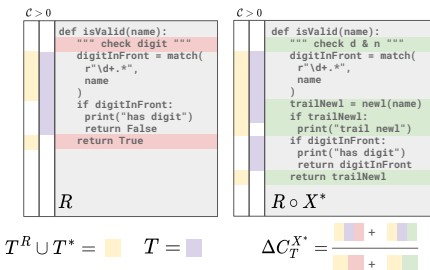

$$\mathcal{X}^*_r = \{l \in X_r \mid \mathcal{C}^R_{T^R}(l) + \mathcal{C}^R_{T^*}(l) > 0\}$$
$$\mathcal{X}^*_a = \{l \in X_a \mid \mathcal{C}^{R \circ X}_{T^R}(l) + \mathcal{C}^{R \circ X}_{T^*}(l) > 0\}$$

where $X_r$ and $X_a$ are the lines added and removed by patch $X$, respectively, and $T^*$ are the golden tests. Finally, we obtain the change coverage of the generated tests $T$ as

Figure 3: Illustration of change coverage $\Delta\mathcal{C}$ of the generated tests $T$, given the test suite $T^R$ of the original code base $R$, the golden patch $X^*$, and the golden tests $T^*$.

$$\Delta\mathcal{C}^X_T = \frac{|\{l \in \mathcal{X}^*_r \mid \mathcal{C}^R_{T^R \cup T}(l) > \mathcal{C}^R_{T^R}(l)\}| + |\{l \in \mathcal{X}^*_a \mid \mathcal{C}^{R \circ X}_{T^R \cup T}(l) > \mathcal{C}^{R \circ X}_{T^R}(l)\}|}{|\mathcal{X}^*_r| + |\mathcal{X}^*_a|}.$$

Where $X$ and $T$ are clear from context, we drop them for notational clarity. If none of the lines modified by the golden patch $X$ are executed by any test, i.e., $|\mathcal{X}^*_r| + |\mathcal{X}^*_a| = 0$, we exclude this instance from our coverage analysis (1% of cases).

**Patch Well-Formedness**   Many LLMs struggle to generate well-formed code patch files (Jimenez et al., 2023) and the methods we investigate employ different approaches to mitigate this issue. To assess them, we additionally measure the patch applicability $\mathcal{W}$ as the portion of instances for which a well-formed patch was generated. We define $\mathcal{W}$ as the portion of instances for which the generated patch $X$ can be applied to the original codebase $R$ without errors. Since well-formedness is necessary for any test to be executed, it always exceeds $\mathcal{S}$, $F \to P$, and related rates.

## 4   Automatic Test Generation

We first discuss how the test generation task differs from code repair, before introducing a novel code diff format based on these insights that is optimized for fault tolerance. Finally, we propose a range of test generation methods based on directly querying LLMs and leveraging Code Agents.

### 4.1   Test Generation vs Code Repair

Automatic test generation is closely related to code repair: Instead of predicting a patch $X$ that fixes the described issue and is then evaluated using a golden test $T^*$, we aim to predict reproducing tests $T$ which are then evaluated on both the original state of the codebase $R$ and the state after applying the golden code patch $X^*$. However, there are some key differences between the two tasks: First, adapting an existing test suite to reproduce an issue typically only requires adding new tests. Concretely, 71% of golden tests in SWT-BENCH only add new test functions, with another 28% modifying existing functions, and only 1% removing functions. Second, testing permits and requires a more granular analysis. While fixed code is either correct and passes all test cases or incorrect when failing any of them, generated tests can be correct but irrelevant to the issue ($P \to P$), call relevant code but fail to expose the precise bug (increase in coverage), reproduce the issue with varying comprehensiveness on edge cases ($F \to P$, with varying coverage), or fail in different ways.

```
1  -- demo/file.py
2  +++ demo/file.py
3  @@-4,5 +4,5 @@
4  def test_euclidean(a, b):
5  -    assert euclidean(1, 0) == 1
6  +    assert euclidean(100, 10) == 10
7      assert euclidean(1, 1) == 1
```

```
1  demo/file.py
2  rewrite
3  1
4  def test_euclidean(a, b):
5      assert euclidean(100, 10) == 10
6      assert euclidean(1, 1) == 1
7  end diff
```

Figure 4: Comparison of the default unified diff format (left) and our fault-tolerant version (right).

## 4.2  A Code Diff Format for Automatic Test Generation

Code changes are typically represented in the unified diff format, i.e., in the git patch and diff format. While using this format to represent code changes is both precise and human-readable, it is very sensitive to misspecifications, requiring, e.g., the exact line numbers of code changes to be specified and specific code snippets (including all to-be-changed lines) to be repeated verbatim. As a result, many LLMs struggle to produce well-formed patch files (Jimenez et al., 2023). Even when loosening the strict diff requirements and fuzzy-matching the generated diff to a best-fit part of the code, GPT-4 only succeeded in $48\%$ of cases, resulting in only $10$ correctly reproduced issues.

To alleviate this issue, we propose an adjusted patch format optimized for LLM generation that is easier to adhere to and more robust. Specifically, our custom diff format allows entire functions or classes to be inserted, replaced, or deleted, given the full function or class definition and (fault-tolerant) location in the code. We show an example in Fig. 4, comparing it to the unified diff format. Based on whether the model wants to rewrite an existing function or insert a new function, the provided code is then substituted or inserted at the code location. This format is particularly well suited for test generation which usually only requires adding test functions. We provide a more formal description of this format in App. A and demonstrate its effectiveness in §5.

## 4.3  Direct LLM Generation of Tests

We consider four baselines for test generation: Direct zero-shot prompting with the unified patch format (ZEROSHOT), zero-shot prompting with our novel patch format (ZEROSHOTPLUS), selecting the best out of 5 patches using an oracle (PASS@5), and the state-of-the-art test generation method, LIBRO (Kang et al., 2023), which uses a range of heuristics to pick the most promising among multiple generated tests. In all methods, the LLM is instructed to add tests to reproduce and cover the described issue in the codebase. We describe these methods below, deferring further details to App. E.

ZEROSHOT prompts the model with the issue description, a subset of the codebase retrieved using BM-25 (Robertson & Zaragoza, 2009), and instructions to generate a patch file in unified diff format. This method corresponds to the LLM-only baseline in SWE-BENCH (Jimenez et al., 2023).

ZEROSHOTPLUS is similar to ZEROSHOT but leverages our custom diff format, discussed in §4.2, which is optimized for LLMs and robustness to minor specification errors.

PASS@5 uses our ZEROSHOTPLUS prompting scheme to generate 5 proposal tests and then uses an oracle to pick the best one. While this is of course not practical in a real-world setting, it allows us to assess the potential of the LLM to generate good test cases given an effective selection mechanism.

LIBRO (Kang et al., 2023), is the current state-of-the-art for LLM-based test generation. Similar to PASS@5 it generates multiple proposal tests using ZEROSHOTPLUS prompting. However, instead of using an oracle, it combines multiple heuristics to select the best test cases. In particular, it runs all generated tests and then selects the one inducing an error that is most similar to the problem description. This permits not only checking whether a generated diff is well-formed and the proposed test fails on the original codebase but also selecting the most relevant test case. As LIBRO was originally proposed for Java, we adapt it to our Python setting, as detailed in App. B.

## 4.4  Code Agents for Test Generation

LLM-based agents are systems that take actions based on LLM-generated text, providing tools to observe and interact with their environment over multiple turns and preserve some state across these

turns. In the case of Code Agents, they can typically search, read, and edit code using an agent computer interface (ACI) (Yang et al., 2024). Recent work has shown that such Code Agents are particularly effective for complex repository-level code synthesis and repair tasks, outperforming unaided LLMs by a significant margin (Bouzenia et al., 2024b; OpenDevin, 2024; Zhang et al., 2024; Yang et al., 2024; Tao et al., 2024). In this work, we leverage Code Agents for automatic test generation by adjusting their instructions. Specifically, we adapt SWE-AGENT (Yang et al., 2024), AIDER (Aider, 2024), and AUTOCODEROVER (Zhang et al., 2024).

SWE-AGENT (Yang et al., 2024) provides the LLM direct access to (augmented) command line tools and processes the output of these tools to be more easily parseable by an LLM. In particular, they provide special tools for searching, viewing, and editing files. Beyond initial instructions, they provide little guardrails or structure for the LLM and let it interact with a limited shell environment.

AIDER (Aider, 2024) performs a repository indexing step to guide file selection and then includes all selected files in the next prompts. Further, model-generated summaries and reflections on previous actions are leveraged to augment the context. Before an edit is applied, it undergoes validation via static analysis and repository test cases using project-specific evaluation harnesses.

AUTOCODEROVER (Zhang et al., 2024) separates the code repair task into two distinct stages. In the first stage, the LLM is tasked with collecting all required context for the task at hand. To this end, it is equipped with a range of advanced code search and navigation tools, allowing it, e.g., to retrieve class signatures, function definitions, or surrounding code snippets. Once the LLM believes it has gathered sufficient context, it proceeds to the second stage. There, the LLM is tasked with generating the actual code patch in a single step, retrying only if the patch can not be applied.

**Adapting Code Agents for Test Generation** As SWE-AGENT, AIDER, and AUTOCODEROVER were designed for program repair, we adapt their system and instruction prompts to focus on creating high-quality test cases. We find that the underlying LLMs are capable of following these changed instructions and successfully generate test cases for up to $87\%$ of issues. Typically, the instruction changes were as simple as replacing phrases like "solve the issue" with "create unit tests that cover the issue". We provide a more detailed description of the used prompts in App. E.

We experiment with instructing SWE-AGENT explicitly to execute the generated test cases before submitting them. We call this variant SWE-AGENT+ and find that this increases the success rate $\mathcal{S}$ from $15.9\%$ to $18.5\%$ (see Table 2). Note we do not provide any information on *how* to run the tests. This contrasts the LIBRO setting, in which the test execution commands are assumed to be known.

## 5 Experimental Evaluation

We leverage SWT-BENCH to compare the performance of different test generation methods and underlying LLMs (§5.2), their relation with the code repair setting (§5.3), and the impact of instance characteristics (§5.4). We further explore hyperparameter ablations in App. C.

### 5.1 Experimental Setup

We consider GPT-4 (gpt-4-1106-preview OpenAI 2023), GPT-4o mini (gpt-4o-mini-2024-07-18 OpenAI 2024), Claude 3.0 Haiku (Anthropic, 2023), Claude 3.5 Sonnet (Anthropic, 2024), Mistral Large 2 (Team, 2024b) (served via the Mistral AI API), and Mixtral 7x22b (Team 2024a served by Together AI TogetherAI 2023), as underlying LLMs, using GPT-4 unless indicated otherwise. We sample at temperature $t = 0$ for all zero-shot methods and agents and at $t = 0.7$ for LIBRO and PASS@5. For SWE-AGENT, AIDER, and AUTOCODEROVER, we use their default settings, restricting the number of API calls to 20, reflection steps to 3, and interaction rounds to 10, respectively. For LIBRO we sample 5 tests. Due to budget constraints, we focus our evaluation on SWT-BENCH-LITE. In App. C we explore and justify this choice of hyperparameters in detail.

### 5.2 Automatic Test Generation

**Comparing Test Generation Methods** We compare test generation performance in Table 2 where all methods have access only to the issue description and the original codebase. We observe that using the original git code-diff format, ZEROSHOT only generates well-formed patches

for $48.6\%$ of issues. Using our novel test-specific code-diff format (ZEROSHOTPLUS) boosts this rate to $89.5\%$ yielding an almost 3x increase in success rate ($\mathcal{S}$) to $9.4\%$. While picking the best among five generated tests (PASS@5) even yields an $\mathcal{S}$ of $20.3\%$, the heuristics employed by LIBRO can only convert about half of this gap into an $\mathcal{S}$ of $14.1\%$, still beating AUTOCODEROVER and AIDER which achieve an $\mathcal{S}$ of $9.1\%$ and $12.7\%$ respectively.

SWE-AGENT, however, outperforms LI-BRO at an $\mathcal{S}$ of $15.9\%$, increased to $18.5\%$, when instructed to check its generated tests (SWE-AGENT+). This stronger performance is significant at $p < 0.1\%$. Interestingly, SWE-AGENT produces fewer $F \rightarrow \times$ tests than AIDER and LIBRO despite having much higher applicability and yielding a higher $\mathcal{S}$.

We conclude that general-purpose Code Agents already perform as well as domain-specific test generation methods, with simple test-specific adjustments providing significant improvements.

Table 2: Rate of well-formed patches ($\mathcal{W}$), successful tests ($\mathcal{S}$), potentially reproducing initially failing tests ($F \rightarrow \times$), reproducing fail-to-pass tests ($F \rightarrow P$), and correct but unhelpful pass-to-pass tests ($P \rightarrow P$), in %.

| Method | $\mathcal{W}$ | $\mathcal{S}$ | $F \rightarrow \times$ | $F \rightarrow P$ | $P \rightarrow P$ |
|---|---|---|---|---|---|
| GOLDEN | 100.0 | 100.0 | 100.0 | 100.0 | 11.2 |
| PASS@5 | 93.1 | 20.3 | 62.7 | 22.1 | 7.2 |
| ZEROSHOT | 48.6 | 3.6 | 38.8 | 5.8 | 3.6 |
| ZEROSHOTPLUS | 89.5 | 9.4 | 55.4 | 10.1 | **7.2** |
| LIBRO | **92.0** | **14.1** | **60.1** | **15.2** | **7.2** |
| AUTOCODEROVER | 47.1 | 9.1 | 43.8 | 9.1 | 7.6 |
| AIDER | 66.7 | 12.7 | **57.6** | 17.0 | 8.7 |
| SWE-AGENT | **87.3** | 15.9 | 48.2 | 16.7 | 9.8 |
| SWE-AGENT+ | 85.5 | **18.5** | 46.4 | **19.2** | **10.1** |

**Coverage of Generated Tests**  We analyze the change coverage $\Delta\mathcal{C}$ of the generated tests, i.e., the portion of executable golden patch code that is covered by the generated tests, in Table 3. Across all methods, we observe a significantly higher coverage on successful instances ($\Delta\mathcal{C}^{\mathcal{S}}$), indicating that coverage is indeed correlated with test quality but more granular than $\mathcal{S}$. Interestingly, SWE-AGENT+ achieves notably higher coverage on successful instances than SWE-AGENT highlighting the impact of providing agents with more test-generation-specific tools to identify promising tests. Further, LIBRO achieves lower coverage than most Code Agents, most likely as a consequence of preferring shorter tests.

Table 3: Change Coverage $\Delta\mathcal{C}$ [%] as defined in §3.3 aggregated over all instances, $\mathcal{S}$ instances and non $\mathcal{S}$ instances ($\neg\mathcal{S}$).

| Method | $\Delta\mathcal{C}^{\text{all}}$ | $\Delta\mathcal{C}^{\mathcal{S}}$ | $\Delta\mathcal{C}^{\neg\mathcal{S}}$ |
|---|---|---|---|
| GOLDEN | 72.0 | 72.0 | - |
| PASS@5 | 31.3 | 65.6 | 22.5 |
| ZEROSHOT | 7.6 | 34.9 | 6.6 |
| ZEROSHOTPLUS | 21.5 | **76.7** | 15.7 |
| LIBRO | **23.8** | 64.2 | **17.0** |
| AUTOCODEROVER | 17.9 | 61.3 | 13.6 |
| AIDER | **27.8** | 59.5 | **23.1** |
| SWE-AGENT | 26.5 | 64.7 | 19.1 |
| SWE-AGENT+ | 27.6 | **69.4** | 18.0 |

**Model Effect**  We compare the effect of different underlying LLMs for SWE-AGENT in Table 4. We observe that not only $\mathcal{S}$ but even applicability ($\mathcal{W}$) is highly sensitive to the underlying LLM's performance, with Haiku, GPT-4o mini, and Mixtral achieving significantly lower performance than GPT-4. More capable models like Claude 3.5 Sonnet and Mistral Large 2 perform on par, with the latter even outperforming GPT-4.

Table 4: Comparison of different underlying LLMs for SWE-AGENT, all in %.

| Model | $\mathcal{W}$ | $\mathcal{S}$ | $F \rightarrow \times$ | $\Delta\mathcal{C}^{\text{all}}$ |
|---|---|---|---|---|
| Mistral L. 2 | 76.1 | **16.3** | 51.4 | 23.0 |
| GPT-4 | **87.3** | 15.9 | 48.2 | 26.5 |
| Cl. 3.5 Sonnet | 67.8 | 12.3 | **59.1** | **30.3** |
| GPT-4o mini | 71.0 | 9.8 | 36.2 | 20.9 |
| Cl. 3.0 Haiku | 20.3 | 2.5 | 6.9 | 3.0 |
| Mixtral 8x22B | 3.3 | 0.7 | 1.8 | 0.9 |

## 5.3  Code Repair and Test Generation

**Test Generation for a Given Code Patch**  To assess the effectiveness of automatic test generation at testing specific, provided fixes, we investigate the effect of providing a (possibly incorrect) code patch, the files it changed, and the test file to be modified instead of the files retrieved with BM25, reporting results in

Table 5: Performance of ZEROSHOTPLUS, given the test file to change, none (-), the golden (✓) or an incorrect (✗) code patch, and the files retrieved via BM-25 ($r$), or modified by the golden (✓) or incorrect patch (✗).

| Test File | Mod. Files | Patch | $\mathcal{W}$ | $\mathcal{S}$ | $F \rightarrow \times$ | $\Delta\mathcal{C}^{\text{all}}$ |
|---|---|---|---|---|---|---|
| - | $r$ | - | **87.8** | 8.1 | 52.3 | 12.5 |
| - | ✓ | ✓ | 85.5 | 10.5 | 64.0 | **18.4** |
| - | ✗ | ✗ | 75.0 | 10.5 | 50.0 | 13.2 |
| ✓ | $r$ | - | 76.7 | **15.1** | **59.3** | 17.6 |

Table 5 in %. We use ZEROSHOTPLUS to generate incorrect patches, resampling $\leq 5$ times and excluding instances where we could not generate an incorrect but applicable patch, reducing the sample size to $n = 172$. Providing the test files to change almost doubles $\mathcal{S}$ from $8.1\%$ to $15.1\%$, pulling even with SWE-AGENT. We observe that meanwhile providing a code patch and the files it changed has a much smaller impact, increasing $\mathcal{S}$ only to $10.5\%$ for both the golden patch and an incorrect patch. This highlights the importance of retrieving the correct context for generating relevant tests. Meanwhile, GPT-4 is able to leverage the correct patch, and to improve the coverage increase of the relevant lines by almost $50\%$, from $12.5\%$ to $18.4\%$.

**Filtering Code Fixes with Generated Tests** State-of-the-art code generation methods only resolve around $20\%$ of cases on SWE-BENCH-LITE (Jimenez et al., 2023). Without suitable tests to distinguish correct from incorrect fixes, the overhead from manual testing (Yang et al., 2008) would thus outweigh any benefits from automatic code generation. To address this issue, we use SWE-AGENT to generate both bug fixes and tests, in a similar manner to Chen et al. (2023a). We then filter the generated bug fixes, retaining only those where all generated tests are $F \rightarrow P$ or $P \rightarrow P$. While only achieving $20\%$ recall, this more than doubles the precision of SWE-AGENT to $47.8\%$, making it significantly more practically useful, highlighting the importance of test generation, and opening an avenue to transferring the results from Chen et al. (2023a) towards more complex and realistic scenarios with more expensive inference and evaluation steps.

**Correlation of Test Generation and Code Repair**
We analyze the overlap between solved instances of SWE-BENCH and SWT-BENCH, showing results in Table 6. We observe that the overlap is small for both methods, with no statistical evidence of correlation (p-values of $80.4\%$ and $72.8\%$ for ZEROSHOTPLUS

Table 6: Overlap in solved instances of SWE-BENCH and SWT-BENCH.

|  | SWT | SWE | Overlap | p-Value [%] |
|---|---|---|---|---|
| ZEROSHOTPLUS | 26 | 16 | 1 | 80.4% |
| SWE-AGENT | 44 | 50 | 7 | 72.8% |

and SWE-AGENT, respectively, under the null hypothesis of independence and uniform hardness), indicating that generating tests and fixes are distinct tasks of different difficulties. We explore this relationship in more detail in App. D.

### 5.4 Test Generation Success and Instance Characteristics

**Effect of Issue Description Length**
We investigate the relationship between issue description length and test generation performance in Fig. 5. We observe a general trend that issues with longer descriptions are easier to generate tests for, with all methods achieving a higher $\mathcal{S}$ for longer descriptions, however tending to slightly decrease for very long descriptions. This is likely due to the increased amount of infor-

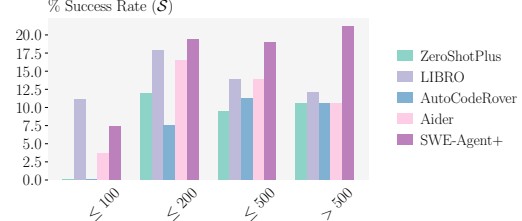

Figure 5: Distribution of success rate ($\mathcal{S}$) across issue description lengths in # tokens

mation available in longer descriptions, while too-long descriptions may contain many distractors and make it difficult to extract relevant information for the LLM. SWE-AGENT+, which actively summarizes context, limiting file content and reducing history, is least sensitive to issue description length, achieving approximately the same $\mathcal{S}$ for all but the shortest lengths.

**Effect of Data Contamination**
As SWT-BENCH is based on historic GitHub issues, they may be contained in the pre-training data of the LLMs we use. To investigate this issue, we conducted an experiment compar-

Table 7: Performance of ZEROSHOTPLUS on PRs before/after GPT-4 knowledge cutoff ($KC = $ 30th April 2023) in %.

| PR created | $n$ | $\mathcal{W}$ | $\mathcal{S}$ | $F \rightarrow \times$ | $F \rightarrow P$ | $P \rightarrow P$ | $\Delta\mathcal{C}^{\text{all}}$ |
|---|---|---|---|---|---|---|---|
| before $KC$ | 83 | 56.6 | 6.0 | 42.2 | 8.4 | 4.8 | 35.9 |
| after $KC$ | 83 | 47.0 | 4.8 | 39.8 | 4.8 | 3.6 | 35.9 |

ing the performance of ZEROSHOTPLUS on all issues created after the Knowledge Cutoff (KC) of GPT-4 (April 2023) to a random subset of the same size of instances created before, and report the results in Table 7. While we observe a small performance difference, we can not confirm its statistical significance ($p \approx 37\%$) due to the low number of samples created after the KC. Further,

all methods in Table 2 use the same underlying LLM and should thus benefit from any potential contamination to a similar degree, allowing for a fair comparison between different methods.

**Method Complimentarity**  We consider four diverse methods from §5.2 and analyze the overlap in the instances for which they are able to generate successful tests. We show the results in Fig. 6. While the best-performing approach, SWE-AGENT+, alone is able to solve 51 instances, the combination of all four approaches is able to solve 87 instances, highlighting the benefit of exploring a variety of approaches for test generation.

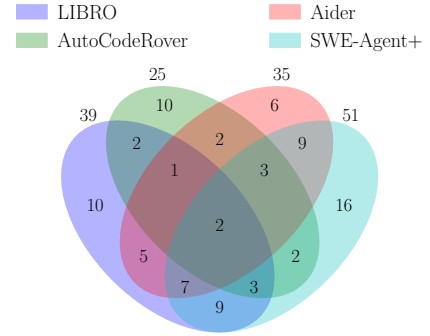

Figure 6: Overlap in instances solved by the four best performing methods.

# 6  Limitations and Future Work

While our novel SWT-BENCH covers a wide range of real-world issues, it has several limitations: It is limited to Python, which may limit the generalizability of our findings to other programming languages. Second, the dataset is based on popular GitHub repositories, which may not be representative of common software development practices and does preclude the generation of a private holdout test set. Finally, the dataset is limited to bug reproduction and issues that can be easily covered by adding test cases and does not measure edge case detection or global coverage increase.

Further, as discussed in §5.4, most issues in SWT-BENCH have been created before the knowledge cutoff of state-of-the-art models, posing a risk for data contamination. One approach to address this issue is to create a rolling version of SWT-BENCH, based only on the most recently created GitHub issues. However, this comes at the cost of direct comparability of results and increased cost for reproducing results for all baselines on a changing evaluation set.

Addressing these limitations would be an interesting direction for future work. As concrete starting points, we found several common errors even in the best performing method SWE-AGENT+ that could be addressed through specialized monitoring: Adding passing tests that do not reproduce the given issue, getting stuck in loops after generating inapplicable edit commands, failing to execute the test environment correctly and adding tests with syntax errors or using invalid variables.

# 7  Conclusion

We proposed SWT-BENCH, a novel benchmark for generating reproducing tests from GitHub issue descriptions and the corresponding code bases. SWT-BENCH leverages the dataset underlying the popular SWE-BENCH which additionally contains a golden patch fixing the described issue. We judge whether a generated test reproduces the described issue by checking whether the test fails before applying this golden patch and succeeds afterward. We measure both the rate of such fail-to-pass tests and the coverage of the golden patch, providing a corresponding evaluation harness. We evaluated a variety of LLM-based test generation methods including Code Agents on SWT-BENCH and found that Code Agents already outperform other approaches with only minor adaptations for the test-generation task. Finally, we demonstrated the great potential of generated tests to serve as a signal for the correctness of code fixes, i.e., we double the precision of Code Agents by filtering the generated patches to only those that cause a previously failing self-generated test to pass.

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

```
1  diff
2  < path or filename >
3  < "rewrite" or "insert" >
4  < line number / EOF / BOF >
5  < function to rewrite or insert >
6  end diff
7  < repeat as necessary >
```

Figure 7: The Custom Diff format for ZEROSHOTPLUS

# A  Formalization of Custom Prompt Format for ZEROSHOTPLUS

We introduce a custom prompt format for language models to aid them with patch generation in the zero-shot setting. The format is visualized in Fig. 7 similar to how it is provided to the language model. A full example of applying the format on two files is part of the full prompt of ZEROSHOT-PLUS in Figs. 11 and 12.

A diff block must start and end with `diff` and `end diff` respectively. The first line inside the block must specify an existing file for rewrites and may point to a new file in the case of insertion. Next, the language model specifies whether it intends to `rewrite` an existing function or `insert` a new function. If no exact match of the function name is found, we employ a fuzzy search using the line number or EOF/BOF as an indicator for where to look for the existing functions. EOF and BOF are particularly useful for inserting new functions. We note that diff blocks can be repeated an arbitrary number of times.

# B  Adapting LIBRO to our Setting

Kang et al. (Kang et al., 2023) originally proposed LIBRO for an evaluation in a pass@k setting. There, it is useful to rank all generated tests to improve performance at $k > 1$. As we only consider pass@1, we drop ranking components irrelevant for the top-1 test in our reimplementation. Further, LIBRO includes heuristics for importing missing dependencies and inserting tests into the correct classes. While effective in Java, this is superfluous for Python, where tests can be added outside classes and dependency imports are (empirically) correctly generated by the LLM. We thus also drop these components.

LIBRO clusters test cases based on whether the generated execution trace matches the issue description. As exact matching is often not applicable for the unstructured issue descriptions, we measure the similarity between the error message and the issue description by extracting the execution trace of the generated test cases and querying the same LLM used for test generation to judge whether they relate to the same issue. Depending on its answer, we obtain two clusters and choose the shortest result of the preferred cluster.

# C  Hyperparameter Ablations

## C.1  Ablation on number of LIBRO samples

We perform an ablation study by varying the number of samples used for the LIBRO baseline. The result is presented in Fig. 8a. LIBRO's performance improves as more samples are considered, however the gains of additional samples are marginal from around the 5 samples we use by default. As shown in section App. G, to enable comparison at similar cost to Code Agents, we settle for 5 samples.

## C.2  Ablation on Interaction Rounds for Code Agents

We analyze the number of interactions required for each Agent to submit a solution and plot the results in Fig. 8b. We observe increasing interaction rounds improve performance until saturation at 5-10 iterations (we use 20 as a default) with the only exception being AutoCodeRover, which still gains performance up to the maximum of 20 iterations we consider.

Table 8: Comparison of ZEROSHOTPLUS for different $T$ on GPT-4 (95% CI, $n = 25$).

| $T$ | $\mathcal{W}$ | $\mathcal{S}$ | $F \rightarrow \times$ | $F \rightarrow P$ | $P \rightarrow P$ | $\Delta \mathcal{C}^{\text{all}}$ |
|---|---|---|---|---|---|---|
| 0.0 | 89.5 | 9.4 | 55.4 | 10.1 | 7.2 | 21.5 |
| 0.2 | $89.3 \pm 0.3$ | $8.8 \pm 0.5$ | $56.2 \pm 0.5$ | $9.6 \pm 0.5$ | $6.0 \pm 0.2$ | $21.8 \pm 0.3$ |
| 0.4 | $89.9 \pm 0.5$ | $10.1 \pm 0.6$ | $56.8 \pm 0.6$ | $10.9 \pm 0.6$ | $6.1 \pm 0.5$ | $22.3 \pm 0.4$ |
| 0.7 | $89.3 \pm 0.4$ | $8.8 \pm 0.4$ | $55.5 \pm 0.8$ | $9.7 \pm 0.5$ | $6.1 \pm 0.5$ | $21.1 \pm 0.6$ |

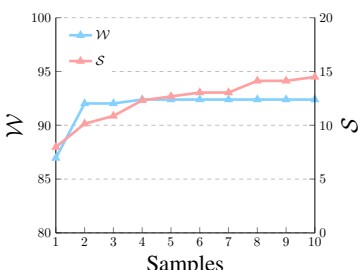 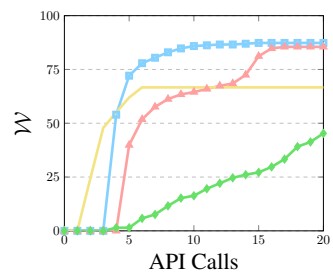 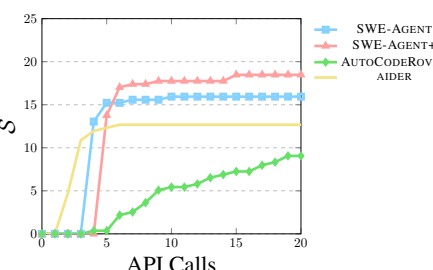

(a) Ablation of $\mathcal{W}$ and $\mathcal{S}$ by number of LIBRO samples.

(b) Ablation of $\mathcal{W}$ and $\mathcal{S}$ by code agent over number of needed API calls until solution submission.

Figure 8: Ablation on the number of samples and API calls for LIBRO and code agents resp.

## C.3 Ablation on Temperature

We run ZEROSHOTPLUS using GPT-4 with 25 samples and analyze the performance variation for a temperature range from greedy decoding ($T = 0$), used for ZEROSHOT, ZEROSHOTPLUS and the agent settings, to $T = 0.7$, the setting used in LIBRO. The results are presented in Table 8. We observe a tendency of decreased performance and increasing variance in all metrics for increasing $T$. Moreover we observe a minimal variability among several runs of the test environment at $T = 0$, however much smaller than the variability due to temperature.

## D   Distribution over Repositories

We compare the success rate of SWE-AGENT for test and fix generation across repositories in Fig. 9. We observe that, while SWE-AGENT obtains similar success rates in both settings in three repositories, success rates vary strongly on most other repositories. Indeed, there are five repositories where test generation fails entirely while code repair fails on three and on only two of these both fail. Manually inspecting instances from the repositories where test generation fails shows a variety

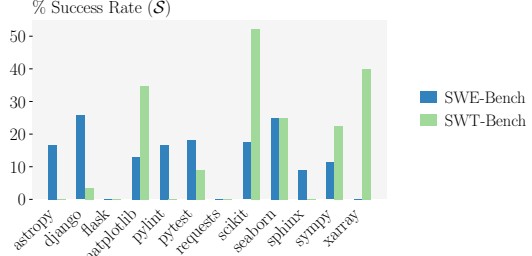

Figure 9: Distribution of success rates across repositories for SWE-AGENT.

of reasons, `astropy` usually features complex computations where unit test creation is difficult, `requests` features a highly complex code base, `flask` has extremely long golden test lengths indicating particularly challenging testing problems. For `pylint` generated tests are all $P \rightarrow P$ making them correct but unhelpful.

## E   Full prompts

**ZEROSHOT, ZEROSHOTPLUS and LIBRO**   The full prompt for ZEROSHOT is displayed in Figs. 10 and 11. The full prompt for ZEROSHOTPLUS and LIBRO is displayed in Figs. 12 and 13. Except for the way we include files, all lines are changed with respect to the setting in SWE-BENCH.

Table 9: Cost of different LLMs running SWE-AGENT on SWT-BENCH Lite in USD

| Model | GPT-4 | Haiku | Mixtral 8x22B | Mistral Large 2 | Sonnet | GPT-4o mini |
|-------|-------|-------|---------------|-----------------|--------|-------------|
| Cost | 290.71 | 10.28 | 67.90 | 211.34 | 263.13 | 8.43 |

Table 10: Cost of running different methods on SWT-BENCH Lite using GPT-4 in USD

| Method | ZEROSHOT | ZEROSHOTPLUS | PASS@5 | LIBRO |
|--------|----------|--------------|--------|-------|
| Cost | 82.13 | 80.70 | 403.65 | 420.14 |
| Method | AIDER | AUTOCODEROVER | SWE-AGENT | SWE-AGENT+ |
| Cost | 256.10 | 368.40 | 290.71 | 478.21 |

This includes in particular the demonstration of the unified diff format on an example. In the setting for Table 5 we add the lines highlighted in **boldface**.

**SWE-AGENT and SWE-AGENT+** The prompt for SWE-AGENT and SWE-AGENT+ is shown in Fig. 14. Changes with respect to the prompt of (Jimenez et al., 2023) are highlighted in **boldface**. The additional changes for SWE-AGENT+ are highlighted in green.

**AIDER** We only minimally adapt the provided evaluation harness for AIDER on SWE-BENCH[1]. In this harness, AIDER is provided with a single initial user prompt based on the user issue, while the entire agent workflow remains unchanged. We provide the entire prompt in Fig. 16 and highlight our change in **boldface**.

**AUTOCODEROVER** AUTOCODEROVER (Zhang et al., 2024) leverages a number of prompts that are provided to the model in different phases of the code/test generation process. We adapt the key prompts and display them in Fig. 15. Changes are highlighted in **boldface**. Further, we change every occurrence of "bug location" in the original prompts to "relevant location". We further add a function to the ACI that allows inserting code in new files and fetching the entire code (capped at the first 100 lines) of any file.

# F   Licenses of used Code

We adapt code from the following projects in our work and include the respective licenses:

1. SWE-BENCH (Jimenez et al., 2023): MIT License
2. SWE-AGENT (Yang et al., 2024): MIT License
3. AIDER (Aider, 2024): Apache License 2.0
4. AUTOCODEROVER (Zhang et al., 2024): GNU General Public License

For all licenses of the repositories used in SWT-BENCH, we refer to Jiminez et al. (Jimenez et al., 2023), which contains a detailed list with licenses for each repository included.

# G   Computational cost

There is cost to both running inference on Language Models and on evaluation their predictions on the test suites of the repositories. Since the evaluation can be performed on a consumer grade machine in reasonable time, we focus on the cost inferred from LLM inference. We report the cost for each setting in Tables 9 and 10, displaying the average cost of a full inference on SWT-BENCH Lite for each model and method. The difference between cost of PASS@5 and LIBRO is just the additional filtering step incurred by LIBRO.

---

[1]https://github.com/paul-gauthier/aider-swe-bench

Table 11: Average execution time $t$ per instance

| Method | ZEROSHOTPLUS | LIBRO | SWE-AGENT | SWE-AGENT+ | AUTOCODEROVER |
|--------|--------------|-------|-----------|------------|---------------|
| $t$ | 12.6s | 2m52s | 3m42s | 4m25s | 5m1s |

## H Execution times

We run the different methods from Table 2 on 5 instances and compute the average execution time. For all LLMs we consider, part of the execution time is directly related to the number of tokens digested and generated (see Table 10). For methods that require interaction with an execution environment however, time is usually dominated by setting up such an environment in a clean and reproducible manner (i.e. dockerized). We list results on execution times in Table 11 and observe that all methods except zero-shot inference take between 3-5 minutes per instance, where we can observe a small trade off due to many-turn interactions in Code Agents versus single-shot execution in LIBRO. Given these small differences however, we believe execution time to be of limited practical relevance as issues can be processed in the background, similar to continuous integration, in response to raised user issues

```
 1  The following text contains a user issue (in <issue/> brackets) posted at a repository.
        Further, you are provided with file contents of several files in the repository that
        contain relevant code (in  brackets). It may be necessary to use code from
        third party dependencies or files not contained in the attached documents however.
        Your task is to identify the issue and implement a test case that verifies a
        proposed solution to this issue. More details at the end of this text.
 2
 3  <issue>
 4  user issue comes here
 5  </issue>
 6
 7  retrieval results or oracle files come here
 8
 9  Please generate test cases that check whether an implemented solution
10  resolves the issue of the user (at the top, within <issue/> brackets).
11  Present the test cases in unified diff formatting.
12
13  The general format of a diff is the unified output format, described as follows.
14  The unified output format starts with a two-line header, which looks like this:
15
16  --- from-file
17  +++ to-file
18
19  Next come one or more hunks of differences; each hunk shows one area where the files
        differ. Unified format hunks look like this:
20
21  @@ from-file-line-numbers to-file-line-numbers @@
22   line-from-either-file
23   line-from-either-file
24
25  If a hunk contains just one line, only its start line number appears. Otherwise its line
        numbers look like 'start,count'. An empty hunk is considered to start at the line
        that follows the hunk.
26
27  If a hunk and its context contain two or more lines, its line numbers look like 'start,
        count'. Otherwise only its end line number appears. An empty hunk is considered to
        end at the line that precedes the hunk.
28
29  The lines common to both files begin with a space character. The lines that actually
        differ between the two files have one of the following indicator characters in the
        left print column:
30
31  '+' A line was added here to the first file.
32  '-' A line was removed here from the first file.
33
34  Insertion can only be done at the end or beginning of the file, indicated by EOF or BOF
        respectively.
35
36  As an example for a diff, consider the following two versions of the same file, once
        before and once after a change.
37  The original version of the file was as follows.
38  [start of demo/test_file.py]
39  1 def test_euclidean(a, b):
40  2     assert euclidean(0, 0) == 0
41  3     assert euclidean(0, 1) == 1
42  4     assert euclidean(1, 0) == 1
43  5     assert euclidean(1, 1) == 1
44  6
45  7 @pytest.mark.parametrize("a, b, expected", [(0, 0, 0), (0, 1, 1), (1, 0, 1), (1, 1, 1)
        ])
46  8 def test_gcd(a, b):
47  9     assert gcd(a, b) == expected
48  10
49  [end of demo/file.py]
```

Figure 10: Part 1 of the Prompt for ZEROSHOT on SWT-BENCH

```
1
2  The diff for fix in function euclidean and adds the function gcd is as follows.
3  This diff changes the first file into the second file.
4  ```diff
5  --- a/demo/file.py
6  +++ a/demo/file.py
7  @@ -4,4 +4,5 @@
8      assert euclidean(1, 0) == 1
9      assert euclidean(1, 1) == 1
10 +    assert euclidean(100, 10) == 10
11
12  @pytest.mark.parametrize("a, b, expected", [(0, 0, 0), (0, 1, 1), (1, 0, 1), (1, 1, 1)
        ])
13 @@ -9,2 +10,6 @@
14     assert gcd(a, b) == expected
15
16 +@pytest.mark.parametrize("a, b, expected", [(0, 0, 0), (0, 1, 1), (1, 0, 1), (1, 1, 1),
        (100, 10, 10)])
17 +def test_lcm(a, b):
18 +    assert lcm(a, b) == expected
19 +
20  ```
21
22  The new version of the file is as follows.
23  [start of demo/file.py]
24  1 def test_euclidean(a, b):
25  2     assert euclidean(0, 0) == 0
26  3     assert euclidean(0, 1) == 1
27  4     assert euclidean(1, 0) == 1
28  5     assert euclidean(1, 1) == 1
29  6     assert euclidean(100, 10) == 10
30  7
31  8 @pytest.mark.parametrize("a, b, expected", [(0, 0, 0), (0, 1, 1), (1, 0, 1), (1, 1, 1)
        ])
32  9 def test_gcd(a, b):
33  10    assert gcd(a, b) == expected
34  11
35  12 @pytest.mark.parametrize("a, b, expected", [(0, 0, 0), (0, 1, 1), (1, 0, 1), (1, 1,
        1), (100, 10, 10)])
36  13 def test_lcm(a, b):
37  14    assert lcm(a, b) == expected
38  15
39  [end of demo/file.py]
40
41  As you can see, you need to indicate the approximate line numbers, function name and the
        path and file name you want to change,
42  but there can be as many independent blocks of changes as you need. You may also apply
        changes to several files.
43  Apply as much reasoning as you please and see necessary. The format of the solution is
        fixed and has to follow the custom diff format.
44  Make sure to implement only test cases and don't try to fix the issue itself.
```

Figure 11: Part 2 of the Prompt for ZEROSHOT on SWT-BENCH

```
 1  The following text contains a user issue (in <issue/> brackets) posted at a repository.
        Further, you are provided with file contents of several files in the repository that
         contain relevant code (in  brackets). It may be necessary to use code from
        third party dependencies or files not contained in the attached documents however.
        Your task is to identify the issue and implement a test case that verifies a
        proposed solution to this issue. More details at the end of this text.
 2
 3  <issue>
 4  user issue comes here
 5  </issue>
 6
 7  The following patch has been proposed to fix the issue described in the user issue (in
        <issue/> brackets).The patch might give you a hint on how to write a covering test
        for the issue, but you should not assume that the patch is correct.It might be that
        the provided patch is not correct, so your test should check whether the patch
        resolves the issue.<patch>proposed patch</patch>
 8
 9  retrieval results or oracle files come here
10
11  Please generate test cases that check whether an implemented solution
12  resolves the issue of the user (at the top, within <issue/> brackets).
13  Present the test cases as a diff (custom format, explained below).
14
15  The general format of a diff is as follows.
16  ```custom-diff
17  diff
18  <path/filename>
19  < "rewrite" or "insert" >
20  < rough line number / EOF / BOF >
21  < insert function that should be added or rewritten >
22  end diff
23  < repeat blocks of diff as necessary >
24  ```
25  Insertion can only be done at the end or beginning of the file, indicated by EOF or BOF
        respectively.
26
27  As an example for a diff, consider the following two versions of the same file, once
        before and once after a change.
28  The original version of the file was as follows.
29  [start of demo/test_file.py]
30  1 def test_euclidean(a, b):
31  2     assert euclidean(0, 0) == 0
32  3     assert euclidean(0, 1) == 1
33  4     assert euclidean(1, 0) == 1
34  5     assert euclidean(1, 1) == 1
35  6
36  7 @pytest.mark.parametrize("a, b, expected", [(0, 0, 0), (0, 1, 1), (1, 0, 1), (1, 1, 1)
        ])
37  8 def test_gcd(a, b):
38  9     assert gcd(a, b) == expected
39  10
40  [end of demo/file.py]
41  ```
```

Figure 12: Part 1 of the Prompt for ZEROSHOTPLUS on SWT-BENCH

```
 1  The diff for fix in function euclidean and adds the function gcd is as follows.
 2  This diff changes the first file into the second file.
 3  ```custom-diff
 4  diff
 5  demo/file.py
 6  rewrite
 7  1
 8  def test_euclidean(a, b):
 9      assert euclidean(0, 0) == 0
10      assert euclidean(0, 1) == 1
11      assert euclidean(1, 0) == 1
12      assert euclidean(1, 1) == 1
13      assert euclidean(100, 10) == 10
14  end diff
15  diff
16  demo/file.py
17  insert
18  EOF
19  @ pytest.mark.parametrize("a, b, expected", [(0, 0, 0), (0, 1, 1), (1, 0, 1), (1, 1, 1),
        (100, 10, 10)])
20  def test_lcm(a, b):
21      assert lcm(a, b) == expected
22  end diff
23
24  The new version of the file is as follows.
25  [start of demo/file.py]
26  1 def test_euclidean(a, b):
27  2     assert euclidean(0, 0) == 0
28  3     assert euclidean(0, 1) == 1
29  4     assert euclidean(1, 0) == 1
30  5     assert euclidean(1, 1) == 1
31  6     assert euclidean(100, 10) == 10
32  7
33  8 @pytest.mark.parametrize("a, b, expected", [(0, 0, 0), (0, 1, 1), (1, 0, 1), (1, 1, 1)
      ])
34  9 def test_gcd(a, b):
35  10     assert gcd(a, b) == expected
36  11
37  12 @pytest.mark.parametrize("a, b, expected", [(0, 0, 0), (0, 1, 1), (1, 0, 1), (1, 1,
      1), (100, 10, 10)])
38  13 def test_lcm(a, b):
39  14     assert lcm(a, b) == expected
40  15
41  [end of demo/file.py]
42
43  As you can see, you need to indicate the approximate line numbers, function name and the
        path and file name you want to change,
44  but there can be as many independent blocks of changes as you need. You may also apply
      changes to several files.
45  Apply as much reasoning as you please and see necessary. The format of the solution is
      fixed and has to follow the custom diff format.
46  Make sure to implement only test cases and don't try to fix the issue itself.
```

Figure 13: Part 2 of the Prompt for ZEROSHOTPLUS on SWT-BENCH

```
 1  We have received following issue within our repository. Here's the issue text:
 2  ISSUE:
 3  {issue}
 4
 5  INSTRUCTIONS:
 6  Now, you're going to create unit tests that cover the issue.  In other words, you should
        write unit tests that fail in the current state of the repositorybut will pass when
        the issue has been resolved.  Essentially, you'll want to write a unit test that
        reproduces the described issue.
 7  Your terminal session has started and you're in the repository's root directory. You can
         use any bash commands or the special interface to help you. Edit all the files you
         need to and run any checks or tests that you want.
 8  Remember, YOU CAN ONLY ENTER ONE COMMAND AT A TIME. You should always wait for feedback
        after every command.
 9  When you're satisfied with all of the changes you've made, you can submit your changes
        to the code base by simply running the submit command.
10  Note however that you cannot use any interactive session commands (e.g. python, vim) in
        this environment, but you can write scripts and run them. E.g. you can write a
        python script and then run it with `python <script_name>.py`.
11
12  NOTE ABOUT THE EDIT COMMAND: Indentation really matters! When editing a file, make sure
        to insert appropriate indentation before each line!
13
14  IMPORTANT TIPS:
15  1. Always start by trying to replicate the bug that the issues discusses.
16      If the issue includes code for reproducing the bug, we recommend that you re-
            implement that in your environment, and run it to make sure you can reproduce
            the bug.
17      Then start trying to fix it.
18      When you think you've fixed the bug, re-run the bug reproduction script to make sure
            that the bug has indeed been fixed.
19
20      If the bug reproduction script does not print anything when it successfully runs, we
            recommend adding a print("Script completed successfully, no errors.") command
            at the end of the file,
21      so that you can be sure that the script indeed ran fine all the way through.
22
23  2. If you run a command and it doesn't work, try running a different command. A command
        that did not work once will not work the second time unless you modify it!
24
25  3. If you open a file and need to get to an area around a specific line that is not in
        the first 100 lines, say line 583, don't just use the scroll_down command multiple
        times. Instead, use the goto 583 command. It's much quicker.
26
27  4. If the bug reproduction script requires inputting/reading a specific file, such as
        buggy-input.png, and you'd like to understand how to input that file, conduct a
        search in the existing repo code, to see whether someone else has already done that.
         Do this by running the command: find_file "buggy-input.png" If that doesn't work,
        use the linux 'find' command.
28
29  5. Always make sure to look at the currently open file and the current working directory
        (which appears right after the currently open file). The currently open file might
        be in a different directory than the working directory! Note that some commands,
        such as 'create', open files, so they might change the current  open file.
30
31  6. When editing files, it is easy to accidentally specify a wrong line number or to
        write code with incorrect indentation. Always check the code after you issue an edit
         to make sure that it reflects what you wanted to accomplish. If it didn't, issue
        another command to fix it.
32
33  7.  After having applied your changes and before submitting, make sure to run pytest and
         check if the code *fails* as expected due to the issue description.  If it doesn't,
         revisit your code changes and adapt them accordingly.
```

Figure 14: The Prompt for SWE-AGENT on SWT-BENCH

```
1  You are a software developer maintaining a large project.
2  You are working on an issue submitted to your project.
3  The issue contains a description marked between <issue> and </issue>.
4  Your task is to invoke a few search API calls to gather information about relevant code
       lines, then write unit tests to capture the described behaviour in the
       issue.Ideally, the unit tests should fail before the bug is fixed or the requested
       feature is added, and pass after.Note you are not trying to solve the bug itself,
       but just capture the behaviour described in the issue by creating appropriate test
       cases.
```

```
1  You are a software developer maintaining a large project.
2  You are working on an issue submitted to your project.
3  The issue contains a description marked between <issue> and </issue>.
4  You ultimate goal is to write one or more unit tests that capture this issue.Ideally,
       the unit tests should fail before the bug is fixed or the requested feature is
       added, and pass after.Note you are not trying to solve the bug itself, but just
       capture the behaviour described in the issue by creating appropriate test cases.
```

```
1  Write one or more unit tests for the issue, based on the retrieved context.
2
3  You can import necessary libraries.
4
5
6  Return the tests as patch in the format below.
7
8  Within `<file></file>`, replace `...` with actual file path.
9
10 Within `<original></original>`, replace `...` with the original code snippet from the
      program.
11
12 Within `<patched></patched>`, replace `...` with the fixed version of the original code.
       When adding orignal code and updated code, pay attention to indentation, as the
       code is in Python.
13 You can write multiple modifications if needed.
14
15 ```
16 # modification 1
17 <file>...</file>
18 <original>...</original>
19 <patched>...</patched>
20
21 # modification 2
22 <file>...</file>
23 <original>...</original>
24 <patched>...</patched>
25
26 # modification 3
27 ...
28 ```
```

Figure 15: The Prompt for AUTOCODEROVER on SWT-BENCH

```
 1  Below is a real GitHub issue from a popular GitHub repository.
 2  The issue was filed some time ago.
 3  The repo has been checked out at the commit that existed at the moment the issue was
       filed.
 4  If you are already familiar with this repo, be cautious!
 5  You are working with an old version of the repo!
 6  Filenames, directory names, file contents, etc may be different than what you're used to
       .
 7
 8  Propose changes to update the repo to reproduce the problem below.
 9  You're going to create unit tests that cover the issue.  In other words, you should
       write unit tests that fail in the current state of the repository
10  but will pass when the issue has been resolved.  Essentially, you'll want to write a
       unit test that reproduces the described issue.
11
12
13  {issue}
```

Figure 16: The Prompt for AIDER on SWT-BENCH

