# OpenReview forum: "SWT-Bench: Testing and Validating Real-World Bug-Fixes with Code Agents"
_NeurIPS.cc/2024/Conference — NeurIPS 2024 poster_

### Official Review · Reviewer_ueFW · 2024-07-03

**Soundness:** 2
**Presentation:** 1
**Contribution:** 2
**Rating:** 3
**Confidence:** 4

**Summary:**

In this paper, the authors propose a novel benchmark for evaluating the capability of LLMs and Code Agents in generating test cases from GitHub issues. The benchmark includes real-world issues, patches, and golden tests from popular Python repositories. The authors claim that Code Agents outperform traditional test generation methods and highlight the potential of these agents in improving software quality and developer productivity.

**Strengths:**

The paper proposes a new benchmark for test generation, addressing a gap in current research.

The approach leverages real-world data from GitHub, making the benchmark relevant and practical.

**Weaknesses:**

The data contamination problem. The proposed benchmark is collected from GitHub, which might have been exposed to the pretraining corpus of LLMs.

The authors overlook substantial existing work on test case generation using LLMs.

There is no discussion on the long-term relevance and maintenance of the benchmark, which is crucial for its sustained utility.

The benchmark is focused solely on Python, ignoring other programming languages.

The experiments are conducted on only three LLMs (GPT-4, Claude-3, Mixtral 7x22B), which is insufficient for a comprehensive evaluation of the benchmark.

The paper fails to address the broader societal impact.

**Questions:**

How does the benchmark ensure that it does not include tasks already present in pre-trained DNN datasets?

Why were significant existing works on LLM-based test generation not included in the paper?

What measures are in place to ensure the long-term relevance and maintenance of the SWT-BENCH benchmark?

How might the results differ if the benchmark included tasks in programming languages other than Python?

Can the authors discuss the potential societal impacts of the widespread adoption of Code Agents in software testing, particularly concerning developer employment?


Please see the above to clarify any misunderstandings and add additional results.

**Limitations:**

The paper lacks discussion on various limitations, including the limitations of the metrics used and potential data contamination across models.

---

> ### Author Rebuttal · Authors · 2024-08-07
>
> We thank Reviewer ueFW for their critical perspective and for raising interesting questions, which we address below.
>
> **Can you discuss the effect of possible data contamination on your results and how it might be mitigated?**
> We thank the reviewer for raising this point. In short, we see no statistically significant difference in performance (p=37%) for issues from before and after the data cutoff and thus conclude that contamination is not affecting our results significantly. Please see the main response for more details.
>
> **Why were significant existing works on LLM-based test generation not included in the paper?**
> We discuss automated test generation (including using LLMs) in our related work (Lines 71-75). Due to space constraints, this discussion focuses on the most important related works and, e.g., omits [1,2,3,4] which are not applicable to our setting as they only consider coverage and generating passing tests [1,3,4], require a large number of tests and fixes to be generated [2], and the to be tested code to be specified [3,4]. We are happy to include these in the next revision using the additional space. If the reviewer could provide a list of works they believe to be relevant, we would be happy to review and include them.
>
> **How can the long-term relevance and maintenance of SWT-Bench be ensured?**
> We agree that long-term relevance is an important topic, actively discussed in the community. We believe SWT-Bench is particularly well positioned to ensure long-term relevance. In particular, new instances can be easily created from new repositories as well as new issues in already used repositories. However, such a rolling benchmark also has disadvantages (comparability of scores, cost of reevaluation). Regardless of these aspects, we believe our initial results showing the promise of Code Agents to already be valuable irrespective of any long-term maintenance of SWT-Bench.
>
> **Can you discuss the relevance of your results on Python and how they might generalize to other languages?**
> Please see our main response for a discussion on Python’s relevance and how our approach applies to other languages. Further, it is common in the field to focus on a single language, especially when addressing a new task, be that Java [3,4], Kotlin [1], or Python [2,5].
>
> **Can you evaluate your benchmark on more than three LLMs?**
> We first want to highlight that while we only considered 3 LLMs (the, at the time, best proprietary model (GPT4), a model balancing cost and performance (Claude 3 Haiku), and a strong open-weight model (Mixtral 8x22B)), we consider a wide range of different agents based on the strongest available model. Given the challenging nature and thus low resulting performance even of GPT4, we refrained from considering weaker models. We want to highlight that all other reviewers specifically highlighted the quality of our extensive evaluation. Finally, since the original submission, stronger models have been released and we have conducted additional experiments on GPT4o-mini, Claude 3.5 Sonnet, and Mistral Large 2 using SWE-Agent, reporting results in Table 2 of the attached PDF. We observe that results follow general model capability as expected.
>
> **Can you discuss the potential societal impact of widespread adoption of Code Agents in software testing, particularly concerning developer employment?**
> Testing and bug reproduction are often neglected among professional developers due to a lack of extrinsic and intrinsic motivation [6]. We thus believe a more widespread adaptation of Code Agents in software testing has the potential to not only improve code quality but also developer productivity and satisfaction. Regarding the potential to displace human developers, benchmarks such as SWT-Bench show that current Code Agents are still far from matching or even outperforming human developers. Instead, they show that human supervision is still essential to leverage current Code Agents. Finally, while we believe that benchmarks such as ours can help drive progress toward better AI systems and thus increased automation, we do not believe our work specifically to have an outsized societal impact beyond the general developments in Generative AI. We are happy to include this discussion in the revised version of the paper.
>
> **Can you discuss the various limitations of your work including the metrics used and potential data contamination?**
> We first want to highlight that we discuss a range of limitations in “Section 6: Limitations”. Regarding data contamination, we are happy to include the new results presented in the attached PDF and discussed in the global response in the paper. We would like to ask the reviewer to explain which further “various limitations, including … of the metrics used” they have in mind.
>
> **Conclusion**
> We hope we were able to address the reviewer’s concerns and questions. We would further like to ask the reviewer, where they see flaws in the soundness and presentation of our work given that all other reviewers rate them as good or excellent. Finally, we would like to ask the reviewer to reconsider if their score is justified given the practical relevance and high potential impact they attest our work with weaknesses being shared by most other work in the space.
>
> **References**
> [1] Alshahwan et al. 2024, Automated Unit Test Improvement using Large Language Models at Meta
> [2] Chen et al. 2022, CodeT: Code Generation with Generated Tests
> [3] Schäfer et al. 2023, An Empirical Evaluation of Using Large Language Models for Automated Unit Test Generation
> [4] Chen et al. 2024, ChatUniTest: A Framework for LLM-based Test Generation
> [5] Wang et al. 2024, TESTEVAL: Benchmarking Large Language Models for Test Case Generation
> [6] Straubinger and Fraser 2023, A Survey on What Developers Think About Testing

---

> ### Comment · Reviewer_ueFW · 2024-08-14
> **Response by Reviewer ueFW**
>
> Thank you for the responses, and sorry for the late response due to a flurry of proposals and review dues. I have increased my overall assessment as some of my concerns have been addressed. While others are shown in below.
>
> **Data contamination** Thanks for your experiments for GPT-3-preview 1106, the experiments do not address my concern. At least, I need to mention the data contamination for all models by directly providing previous inference results and dividing them based on the KC time.
>
> **Existing works not included in the paper?** Thanks for your response. But I need to mention that omitting CodeT is crazy.
>
> **Other languages?** The response does not address my concern.
>
> **Other LLMs** I would recommend providing experiments for OpenCodeInterpreter, DeepSeek-Coder, XwinCoder, CodeLlama, WizardCoder, and Starcoder2 family in open-source LLMs for a new benchmark.

---

> > ### Author Response · Authors · 2024-08-14
> > **Reply to Reply to ueFW**
> >
> > We thank the reviewer for acknowledging our rebuttal, engaging in the discussion, and raising their score. We would like to address their comments below.
> >
> > **Can you provide the data contamination results for each model?**
> > Below we report results on SWT-bench Lite for all considered models, split by whether the issue was created before or after the corresponding Knowledge Cutoff (KC). The results correspond to Table 2 in the global attached PDF, comparing the model performances using SWE-Agent. Note that for our previous results on GPT4, we considered the full SWT-bench to ensure a sufficiently large number of samples from after the KC is available. Due to time constraints, this is not possible here and the smaller size of SWT-bench Lite results in much fewer instances being available after the KC. In fact, many of the recent models have a KC later than the latest instance (15 Aug 2023) in the full SWT-Bench (see the table below). For the only model with sufficiently early KC (Mixtral 8x22B), the overall F2P rate is so low that no meaningful comparison is possible (2/190 F2P instances before KC and 0/89 after).
> >
> > | Model                 	| KC          | PR created   |   n | Applicable   |F2P   | Coverage   |
> > |---------------------------|-------------|--------------|-----|--------------|-------|------------|
> > | GPT-4 Preview 1106    	| 30 Apr 2023 | before KC	| 268 | 76.9     	| 16.4  | 18.1   	|
> > |                       	|             | after KC 	|  11 | 54.5     	| 18.2  | 20.0   	|
> > | Mistral Large 2       	| 31 Jan 2024 | before KC	| 279 | 4.3      	| 0.7   | 0.2    	|
> > |                       	|             | after KC 	|   0 | -        	| - 	| -      	|
> > | Claude 3.5 Sonnet     	| 30 Apr 2024 | before KC	| 279 | 60.2     	| 12.2  | 22.1   	|
> > |                       	|             | after KC 	|   0 | -        	| - 	| -      	|
> > | GPT-4o mini  (2024-07-18) | 31 Oct 2023 | before KC	| 279 | 53.8     	| 9.7   | 13.0   	|
> > |                       	|             | after KC 	|   0 | -        	| - 	| -      	|
> > | Claude 3.0 Haiku      	| 31 Aug 2023 | before KC	| 279 | 6.8      	| 2.9   | 1.9    	|
> > |                       	|             | after KC 	|   0 | -        	| - 	| -      	|
> > | Mixtral 8x22B         	| 30 Sep 2021 | before KC	| 190 | 2.6      	| 1.1   | 0.0    	|
> > |                       	|             | after KC 	|  89 | 2.2      	| 0.0   | 1.4    	|
> >
> >
> > **Why did you not discuss CodeT in the paper?**
> > As discussed in detail in our response to Reviewer PhkB, CodeT is not applicable to our setting, and we thus omitted it in our literature review due to space constraints. However, we will make sure to include it with a suitable discussion in the next revision of our paper, using the extra space of the camera ready version.
> >
> > **Why did you not consider popular Open-Source Code Models like CodeLlama?**
> > During our experimentation, we indeed evaluated CodeLlama-70B and WizardCoder-Python-34B. However, we found they were not capable of following the desired diff format or agent instructions. Instead, they produced incoherent, incorrect, and degenerated outputs. As they thus consistently yielded close-to-zero F2P rates, we have excluded them from the reported results. We do not believe that comparing such low F2P rates would be interesting or meaningful, but are happy to include results for these models in the appendix of a revised version. Since both the strongest models (e.g. GPT4) alone (ZeroShotPlus) and slightly weaker models (e.g. Claude 3.0 Haiku) in an agent framework struggle with SWT-Bench, we believe it is most interesting to focus on comparing different agent frameworks using the best available base models.
> >
> > **Conclusion**
> > We hope to have been able to address the reviewers concerns and are looking forward to their response.

---

> > > ### Comment · Reviewer_ueFW · 2024-08-14
> > >
> > > Dear authors
> > >
> > > Thanks for your response.
> > >
> > > I find that the Applicable (pre-KC) for GPT-4-turbo-preview are largely different with post-KC. Can you clarify that?

---

> ### Author Response · Authors · 2024-08-14
> **Reply to Reply to Reply to ueFW**
>
> We thank the reviewer for their quick reply and hope that we could successfully address all points they did not have follow-up question on.
>
> **Effect of Data Contamination on Applicability for GPT4**
> While the applicability is indeed higher for issues created before the KC for GPT4, both F2P rate and Coverage are lower, which are much stronger indicators for memorization. Applicability solely measures whether the LLM can respect the required diff format, while F2P and Coverage measure the correctness of the generated tests. While memorization would lead to a correct test (higher F2P and coverage) it would not necessarily improve applicability as the test would not have been included in the training data in the right diff format. Further, we want to highlight that only 11 samples after the KC were available, making any conclusions based on these results statistically questionable (as highlighted in our previous reply).

---

### Official Review · Reviewer_pxq3 · 2024-07-12

**Soundness:** 3
**Presentation:** 3
**Contribution:** 2
**Rating:** 5
**Confidence:** 4

**Summary:**

This paper introduces SWT-BENCH, a novel benchmark for evaluating automated test generation capabilities of AI models, particularly Code Agents. The authors adapt existing code repair datasets and methods to the task of test generation, proposing new metrics such as fail-to-pass rate and change coverage. Their experiments reveal that Code Agents, originally designed for code repair, outperform methods specifically designed for test generation. The best-performing method, SWE-AGENT+, achieves an 11.1% success rate in generating relevant fail-to-pass tests. The study also demonstrates significant complementarity between different approaches, with an ensemble of methods solving 70% more samples than the best single method. Additionally, the authors show that generated tests can serve as a strong signal for the correctness of proposed code fixes. While the study is limited to Python and may have some selection biases, it suggests that Code Agents have significant potential for automated test generation, potentially improving both software quality and developer productivity.

**Strengths:**

Originality:
- Introduces SWT-BENCH, a novel benchmark for test generation, adapting existing code repair datasets to a new task.
- Proposes the use of Code Agents for test generation.
- Develops new metrics (fail-to-pass rate and change coverage) specifically for evaluating test generation.

Quality:
- Comprehensive evaluation of multiple methods, including baselines and state-of-the-art approaches.
- Rigorous experimental setup with detailed reporting of methodologies and results.
- Great analysis of results, including complementarity between methods and correlation with code repair performance.

Clarity:
- Well-structured paper with clear explanations of complex concepts.
- Effective use of figures and tables to illustrate key points and results.
- Detailed appendices providing full prompts and additional experimental details.

Significance:
- Demonstrates the potential of Code Agents for automated test generation, a critical area in software development.
- Shows that generated tests can effectively validate code fixes, potentially improving code quality processes.
- Provides a new benchmark (SWT-BENCH) that could drive further research in this area.
- Highlights the complementarity of different approaches, suggesting potential for ensemble methods in this domain.

**Weaknesses:**

1. Limited statistical analysis: The authors acknowledge they were unable to perform a full statistical analysis due to computational costs. This limits the confidence in the reported results. It would be great if they can conduct a power analysis to determine the minimum number of runs needed for statistical significance.

2. Lack of error analysis: The paper doesn't provide a detailed analysis of the types of errors made by different methods. Maybe they should categorize and quantify common error types for each method. Provide qualitative examples of generated tests, both successful and unsuccessful. Analyze how error types correlate with issue complexity or repository characteristics.

3. Limited exploration of hyperparameters: The paper doesn't discuss the impact of different hyperparameters on the performance of Code Agents. To improve: Conduct an ablation study on key hyperparameters (e.g., number of interaction rounds, temperature). Provide insights on how to optimize Code Agents specifically for test generation.

4. Insufficient comparison to human performance: The paper lacks a comparison to human-generated tests. To address this: Include a small-scale study with professional developers generating tests for a subset of issues. Compare the quality, coverage, and time efficiency of AI-generated vs. human-generated tests.

5. Narrow focus on Python: The study is limited to Python, potentially limiting generalizability.

**Questions:**

1. Given the computational constraints that prevented a full statistical analysis, could you provide more details on the variability of your results? Even with limited runs, could you estimate confidence intervals for your main findings?

2. The paper lacks a detailed error analysis. Could you provide examples of common failure modes for the different methods, particularly for SWE-AGENT+? How do these failure modes relate to the characteristics of the issues or repositories?

3. How sensitive are the Code Agents' performances to different hyperparameters? Did you explore variations in the number of interaction rounds or temperature settings? If so, what insights can you share about optimizing Code Agents specifically for test generation?

4. The paper doesn't compare AI-generated tests to human-generated ones. Have you considered conducting even a small-scale comparison with professional developers? This could provide valuable context for understanding the practical impact of your methods.

**Limitations:**

1. Limited Statistical Analysis: Due to computational constraints, the authors were unable to perform a full statistical analysis. This limits the confidence in the reported results and makes it difficult to assess the robustness and reproducibility of the findings.

2. Lack of Detailed Error Analysis: The paper doesn't provide an in-depth analysis of the types of errors made by different methods. This limits understanding of where and why the methods fail, which could be crucial for further improvements.

3. Python-Centric Approach: The study focuses exclusively on Python, which may limit the generalizability of the results to other programming languages.

4. Absence of Human Baseline: There's no comparison between AI-generated tests and human-generated tests, making it challenging to assess the practical impact and efficiency gains of these methods in real-world scenarios.

5. Limited Hyperparameter Exploration: The paper doesn't thoroughly explore the impact of different hyperparameters on the performance of Code Agents, potentially missing opportunities for optimization.

6. Lack of Runtime Performance Analysis: While computational costs are mentioned, there's no detailed analysis of execution times for different methods, making it difficult to assess practical trade-offs.

---

> ### Author Rebuttal · Authors · 2024-08-07
>
> We thank Reviewer pxq3 for their detailed review, insightful questions, and helpful suggestions. We are happy to hear they appreciate the originality, quality, clarity, and significance of our work and in particular our comprehensive and rigorous evaluation across methods and models. Below, we address their remaining questions.
>
> **Can you conduct additional experiments to establish statistical significance and/or report confidence intervals?**
> Please see the main response for a detailed answer. In short, we have added experiments and statistical analysis demonstrating a high statistical significance of our results.
>
> **Can you conduct an error analysis, categorizing common errors made by different methods, and provide qualitative examples? Do these error types correlate with issue complexity or repository characteristics?**
> We have conducted a detailed analysis of common errors for our best-performing method SWE-Agent+ and will gladly add the findings to the paper for the camera-ready version together with concrete examples of their occurrence. We find the most common issues of the agent to be adding tests that do not reproduce the issue (either due to unrelated failure or non-failure) and incorrectly classifying the result as relevant, getting stuck in a loop after making inapplicable edits or encountering syntax errors, failing to execute the test environment after test creation, and adding tests with missing fixtures or missing indentation, rendering the test function non-executable. Further, we want to point the reviewer to Section 5.4, where we already investigate correlations of success with repositories, issue description length, and code repair success.
>
> **Can you conduct an ablation study on key hyperparameters of the employed code agents?**
> In the attached PDF, we provide ablation studies on the temperature used for decoding, the number of interaction rounds for various agents, and the Libro sample size (see Table 3 and Figure 1a,b). In agreement with prior work, we consistently observe the best performance for greedy decoding (t=0). We further observe increasing interaction rounds improve performance until saturation at 5-10 iterations (we use 20 as a default) with the only exception being AutoCodeRover, which still gains performance up to the maximum of 20 iterations we consider. Similarly, Libro’s performance improves as more samples are considered, however saturating also around the 5 samples we use by default. We will include these ablations confirming the choices of our hyperparameters in the updated appendix.
>
> **Can you conduct a small-scale study with professional developers to establish a baseline in terms of quality, coverage, and time efficiency?**
> We believe a human study of statistically significant sample size to be infeasible as a baseline given that many of the issues at hand are complex and challenging to understand, with the original underlying issues taking up to a year to resolve. Further, as all ground truth tests were written by humans, we believe that most professional developers should be able to solve this task given sufficient (substantial) time. We want to highlight that such human baselines are uncommon for code benchmarks, even at the much simpler function-synthesis level, due to the above-mentioned substantial time requirements. Finally, we believe that establishing such human baselines, possibly across skill levels, and focusing on simpler benchmarks could be an interesting work on its own and leave it to future work.
>
> **Can you discuss the applicability of your approach beyond Python?**
> Please see our answer in the main response.
>
> **Can you include an analysis of execution times for different methods?**
> For all LLMs we consider, part of the execution time is directly related to the number of tokens digested and generated (see Tables 7 and 8 of the main submission). For methods that require interaction with an execution environment however, time is usually dominated by setting up such an environment in a clean and reproducible manner (i.e. dockerized). We list results on execution times below and observe that all methods except zero-shot inference take between 3-5 minutes per instance, where we can observe a small trade off due to many-turn interactions in Code Agents versus single-shot execution in LIBRO. Given these small differences however, we believe execution time to be of limited practical relevance as issues can be processed in the background, similar to continuous integration, in response to raised user issues.
>
> | Method | Execution Time |
> |---|---|
> | ZeroShotPlus | 12.6s |
> | LIBRO | 2m53s |
> | SWE-Agent | 3m42s |
> | SWE-Agent+ | 4m25s |
> | AutoCodeRover | 5m1s |
>
>
> **Conclusion**
> We hope to have been able to address the reviewer's concerns, in particular regarding the statistical significance and hyperparameter sensitivity of our approach, remain happy to answer their follow-up questions and look forward to their reply.

---

### Official Review · Reviewer_Ky1F · 2024-07-15

**Soundness:** 4
**Presentation:** 4
**Contribution:** 4
**Rating:** 8
**Confidence:** 4

**Summary:**

This paper focuses on automatic test generation using large language models (LLMs) and code agents. The authors introduce a benchmark called SWT-BENCH, which aims to analyze the performance of LLMs and agents in generating unit tests given a task description. The key contributions include:

1) Creating the SWT-BENCH testing benchmark, which contains over 1700 samples from real-world GitHub issues.

2) Benchmarking various LLMs and agent systems. This includes evaluating direct LLM prompting, existing test generation tools (e.g., LIBRO), and adapted code agents (e.g., SWE-AGENT, AUTOCODEROVER).

3) Introducing a new format for LLM-based code editing.

**Strengths:**

1) LLMs are stochastic black boxes by nature, and building experimental testbeds is crucial to advance the field. They can provide training data for the models as well as relevant signals for improvements. I think this benchmark is a very valuable contribution to the field as it will enable more algorithmic and agentic advances on the topic.

2) This benchmark does a good job at testing several methods such as LLMs, agents, and modifications to existing agentic systems.

3) The performance analysis of various methods is quite detailed, and the correlation or lack thereof between methods is very interesting.

**Weaknesses:**

1) The scope of the dataset could be expanded further. In its current form, it only addresses Python codebases with associated GitHub PR issues. The distribution of testing problems encountered by real-world software engineers is likely quite different from those represented in this testbed.

2) A private test set that is not part of the publicly available training data would strengthen the claims that could be made when improvements are observed on this benchmark.

**Questions:**

1) Have you looked into leveraging the failure modes of the generated test cases to improve code repair or code generation tasks?

2) LLMs for code generation seem to be very stochastic by nature. You mention that the failure modes on the benchmark vary depending on the underlying method used. To what extent do you think this variation is explained simply by the stochasticity of the method? For example, if you ran swe-bench 5 times, would that variation disappear?

3) How do you think would your method generalize in a private codebase that isn't available online?

**Limitations:**

The contamination problem with LLMs is, in my opinion, very important and under-discussed. Due to GitHub data being public, a lot of the data already exists in the training data for LLMs, which can render some conclusions less impressive. This is especially true in the code repair and test generation areas. I think it would be very helpful to include some analysis of the contamination problem. This could be done by using a held-out dataset in a similar way to how the benchmark below does it:

https://livecodebench.github.io/

---

> ### Author Rebuttal · Authors · 2024-08-07
>
> We thank Reviewer Ky1F for their detailed review, insightful questions, and helpful suggestions. We are happy to hear they appreciate the potential impact of our work as well as the extensiveness of our empirical evaluation and its analysis. Below we address their remaining questions.
>
>
> **Can the scope of the benchmark be expanded to address other repositories and programming languages?**
> Our approach can be applied to any programming language and can be easily extended to other repositories, including private projects if Github issues and pull requests were used. However, setting up the corresponding execution environments with historically correct dependency versions and parsing of all test results requires some manual effort. Given the already substantial cost of evaluating the whole benchmark, we decide to leave this to future work targeting specific languages or repositories (please also see our answer in the main reply). Further, we want to highlight that most currently used benchmarks such as HumanEval ([1], 164 instances) and the concurrent Software Test benchmark TestEval ([2], 210 instances), focus on much smaller sets of standalone functions in one or few languages, which are much less representative of real-world issues than the popular Github repositories our benchmark is built on.
>
> **Can you add a private test/holdout set to avoid contamination?**
> Having a fully private holdout set is challenging for agent evaluation, as this would require the party hosting this benchmark to execute submitted methods, which can incur substantial costs. Alternatively, the issue descriptions and codebase details would need to be shared with submitters, which would defeat the purpose of a private holdout set. Besides these practical challenges, the code underlying any such holdout set would still be available online, making it challenging to guarantee that it was not used for training. A previously entirely private repository, circumventing the latter issue, is difficult to obtain. Sticking to online code, a rolling benchmark set of recent instances could be used, similar to live code bench. However, this has the issue that reported numbers can not be compared directly and rerunning all methods on the current version would be necessary when evaluating the benchmark. We thus leave this variation to future work.
>
> **Can the failure modes of the test generation be leveraged to improve code repair?**
> We believe that test generation provides a valuable alternative perspective on the capabilities of Code Agents and help drive developments in the field. It can for example unveil repositories where Agents struggle to leverage even the existing tests, highlighting the need for further improvements in Agent tooling. We find several such cases, especially in the repository django, where executing the test suite itself is non-trivial but would benefit Code Agents for validating their changes. Even when test suites are executed, we find that the model often wrongly considers unrelated failures as related to the given issue. Other failures, such as getting stuck in editing loops seem universal and their resolution would benefit both code repair and test generation. We leave a detailed exploration of these directions for future work.
>
> **To what extent does the stochasticity of LLMs explain the variance in failure modes and performance between methods?**
> We first want to highlight that we use greedy decoding for most methods (except Libro and pass@k) which is deterministic up to floating point errors. Further, since the submission, more powerful cheaper models have been released, allowing us to investigate the variance across multiple runs with GPT4o-mini. There, we obtain the results shown in Table 3 of the attached PDF, showing 95% confidence intervals obtained with 5 runs for a range of temperatures. We observe very low variance, which is interestingly dominated by the flakiness of tests (despite execution in the same docker), present even for greedy decoding (temperature 0).
>
> **Can you comment on the generalizability of your results to private codebases that are not available online?**
> Based on manual inspections of agent traces we find that models typically don’t show signs of memorization. In a few cases, the model tries to assess files that do not exist, pointing to either hallucinations or outdated memorization. Overall, we observe that models make use of the provided tooling to navigate and explore the code base. Therefore we conjecture that our results should transfer well to private codebases. We further confirm this by comparing the performance of ZeroShotPlus on all instances created after the training data cutoff of GPT4 to a random subset of instances created before. We observe no statistically significant difference (p-value of 37% using Students t-test).
>
> **Conclusion**
> We hope we were able to address all of the reviewer's remaining questions and remain happy to answer any follow-up questions they might have.
>
> **References**
> [1]  OpenAI 2021, Evaluating Large Language Models Trained on Code
> [2] Wang et al. 2024, TESTEVAL: Benchmarking Large Language Models for Test Case Generation

---

### Official Review · Reviewer_i5fN · 2024-07-18

**Soundness:** 3
**Presentation:** 3
**Contribution:** 2
**Rating:** 4
**Confidence:** 4

**Summary:**

This paper re-purposes SWE-Bench, a previous benchmark on repository-level code generation, to SWT-Bench, a benchmark for test generation by asking LLMs to generate tests to reproduce the issues for various GitHub repos and see if such tests can capture the bugs before the gold-standard patches are applied and whether the generated tests will pass after the patches are applied (i.e., Fail -> Pass). Experiments are conducted with multiple baselines and results show that SWE-Agent, the previously proposed method for code repair, achieves the best performance for test generation.

**Strengths:**

S1. This paper targets on test generation, which is a relatively under-explored area and could be useful for increasing popular coding agents;
S2. This work thoroughly studies the relation between the tasks of code repair and test generation, and show how we can turn a code repair dataset into a test generation dataset using SWE-Bench -> SWT-Bench as an example;
S3. The designed metrics to evaluate the generated tests (i.e., Fail-to-Pass rate, Change Coverage, and Patch Applicability) are quite interesting and reasonable, which could be useful for further work on automatic test generation evaluation.

**Weaknesses:**

I think the main weakness of this paper is over-claiming. While it is a clever idea to re-purpose code repair benchmarks as SWE-Bench into a test generation benchmark, there are quite a lot limitations of this method for the conclusions on the resulting dataset to claim "state of the art software tester". The main purpose of software testing is not to reproduce the issues, but given the specifications, crafting inputs and the constraints on the outputs to makes sure that the code to be tested has the expected behavior given different inputs. To this end, I believe there is a large gap between the ability that SWT-Bench is testing, and the actual test generation ability needed for the job of software testers. To make such a claim, I believe at least a few software testing benchmarks (e.g., https://swtesting.techconf.org) need to be used and the results to be reported.

**Questions:**

Please respond to the concerns in the "Weakness" section.

---

> ### Author Rebuttal · Authors · 2024-08-07
>
> We thank Reviewer i5fN for their review and valuable questions. We are happy to hear they consider the problem we study important and under-explored, our work thorough and interesting and our results impactful. Below, we address their remaining concern.
>
> **Is your paper over-claiming and implying that SWT-Bench measures all abilities of human software testers?**
> While our title may be placative, it only implies that Code Agents perform better than other automated systems on the studied task. In this case, this is the task of reproducing an issue from a description in the form of a test. We fully agree with the reviewer that there is a big gap between the skills SWT-Bench is testing and the full spectrum of abilities required by human software testers. We want to highlight that we do not make any such claims anywhere in our paper. Instead, we explicitly focus on reproducing tests (see e.g. the Conclusion (Line 408) and the Introduction (Line 35)). We will make it even more explicit in the revision that reproducing issues is the only measured skill of multiple software testing skills.
>
> **How important is the measured capability of reproducing issues?**
> While reproducing a described issue or testing a specific functionality is certainly not the sole purpose of software testing, we firmly believe it is a very important one. In particular, it is an aspect of software testing that is difficult to solve without powerful NLP processors such as LLMs. The task of issue reproduction requires translating natural language descriptions of these issues/intended functionalities and into formalized definitions of the desired behavior in the form of unit tests. This is in stark contrast to the task of, i.e increasing coverage of a given (subset of a) codebase, or fuzzing, where the target is already formally defined. Finally, our approach permits test generation for test-driven development, as it can generate tests for unimplemented functionality, i.e. when the raised user issue is a feature request. We thus find it important and promising to study this (relatively under-explored) aspect of software testing.
>
> **Conclusions**
> We hope to have been able to address the reviewer’s concerns regarding the claims made by our paper and would like to respectfully ask them to reconsider their evaluation given their own positive assessment of our work. We are happy to answer follow-up questions and would greatly appreciate it if the reviewer could highlight specific sections where they believe we overclaim.

---

> ### Comment · Reviewer_i5fN · 2024-08-13
> **Thanks for the response**
>
> I'd like to thank the authors for the response.
>
> > While our title may be placative, it only implies that Code Agents perform better than other automated systems on the studied task.
>
> I think this is where the major discrepancy is: the studied task is **GitHub issue reproduction**, which is quite far from what people would normally think of the work of **software testers**. Thus I still think the title is over-claiming and somewhat misleading, and I'm afraid I could not give higher score for this work.
>
> Respectfully,
> Reviewer i5fN

---

> ### Author Response · Authors · 2024-08-13
> **Reply to Reply of i5fN**
>
> We thank the reviewer for their response and engaging in the discussion. We are happy to modify the title of our submission (in accordance with the submission guidelines) to follow the reviewer’s suggestion and avoid over-claiming.
>
> We propose to adjust the title to **Can Code Agents Reproduce Real-World GitHub Issues?**.
>
>  As promised in the initial rebuttal, we will also revise the text to make the scope of our work more explicit. We are happy to answer any followup questions or suggestions they might have.

---

### Official Review · Reviewer_PhkB · 2024-07-23

**Soundness:** 3
**Presentation:** 4
**Contribution:** 3
**Rating:** 7
**Confidence:** 4

**Summary:**

Authors present a new benchmark for the SWE tasks of generating tests corresponding to an issue in a codebase with unit tests. They propose to repurpose the SWE-bench dataset for this task. They also evaluate LLM based prompting and agentic approaches (SWE-Agent and AutoCodeRover) with metrics like code change coverage and fail to pass rate that are formally defined and measured.

**Strengths:**

- This work focuses on generating test cases corresponding to user issues with a codebase with existing unit tests. This problem has been not studied in such detail in my knowledge. Authors have presented this problem with sufficient motivation.
- While authors adapt an existing code fix benchmark (SWE-bench) for their task (SWT-bench), their contributions in the form of task formulation, corresponding new metrics and exhaustive analysis of the benchmark could be key to future research on this topic.
- Authors also adapt popular LLM approaches like SWE-Agent and ACR for their benchmark and show their effectiveness on generating tests. The demonstrated impact automated test generation has on automated code fixing is particularly promising. Their results include exhaustive analysis of the performance of different methods on the proposed benchmark.
- The paper is very well written and clear to follow.

**Weaknesses:**

- Arguably the most significant impact of automatic test generation would be on automated code repair/generation. While authors describe this aspect of their work (Lines 328-334), providing more details and large-scale experiments could significantly strengthen the perceived utility of this work.
- CodeT by Chen et al https://arxiv.org/abs/2207.10397 is a key related work that was missed in the related work.

**Questions:**

- Lines 328-334: You mention about the impact on precision and recall, can you share more details of the method used and the final correctness rate with synthetic tests generated by the Agent?
- What are some future directions you would recommend to improve the performance of LLM Agents on SWT-bench and SWE-bench based on your study? A future work recommendation would greatly help the completeness of the paper.

**Limitations:**

Authors have briefly discussed limitations of their work in the main paper.

---

> ### Author Rebuttal · Authors · 2024-08-07
>
> We thank Reviewer PhkB for their detailed review, insightful questions, and helpful suggestions. We are happy to hear they appreciate the importance and novelty of the problem we study, the importance of our contributions, the extent of our analysis, and the quality of our exposition. Below we address their remaining questions.
>
> **Can you provide more details on how the generated tests improve the precision of code repair agents? And can you conduct more large-scale experiments in this direction**
> Using SWE-Agent, we generate tests and code patches independently for all instances. We then run the generated code patches against the generated tests and retain only those code patches where the tests fail before the patch is applied and succeed after. Evaluating the resulting patches against the ground truth tests shows a precision of 47.5%, significantly higher than the precision of unfiltered patches (19.2%). How to best leverage tests directly for code generation rather than simply selection is an active topic of research with many competing approaches. We therefore leave this as a promising future work item.
>
> **How does your work relate to CodeT by Chen et al.?**
> CodeT focuses on selecting the best implementation(s) from a large set of proposals (100) by generating large numbers of tests and using execution agreement as a quality metric to select correct ones. However, while this approach is effective for function-level, HumanEval-style problems, we focus on significantly more challenging repository-level synthesis. There, neither sampling nor executing such large numbers of different tests and implementations is feasible due to the substantial computational and monetary (when using APIs) cost. In particular, ZeroShot sampling of 100 tests and repairs would cost around $15k for one evaluation of SWT-Bench-Lite. Therefore, we do not compare to CodeT directly. We will include a corresponding discussion in our related work section.
>
> **Can you discuss future directions for improving the performance of LLM Agents on SWT-bench and SWE-bench?**
> We recommend future work to further explore the relationship between patch generation and test generation to improve the performance for both tasks. CodeT [1] is an interesting prior work in this direction. However, it relies on a large number of candidate implementations and tests to find agreements, which is too costly for real-world repositories. Therefore, future work might consider developing more cost-effective approaches.
>
> We have further conducted a detailed analysis of common errors for our best-performing method SWE-Agent+ and will gladly add the findings to the paper for the camera-ready version together with concrete examples of their occurrence. We find the most common issues of the agent to be adding passing tests that do not reproduce the issue, getting stuck in a loop after making inapplicable edits or encountering syntax errors, failing to execute the test environment during test creation, and adding tests with syntax errors. Future work may consider addressing these issues through specialized modules or additional monitoring.
>
> We are happy to include this analysis, discussion and recommendations for future work in the camera-ready version of the submission.
>
> **Conclusion**
> We hope to have been able to address all of the reviewers questions and remain happy to answer any follow up questions they might have.
>
> **References**
> [1] Chen et al. 2022, CodeT: Code Generation with Generated Tests

---

### Author Rebuttal · Authors · 2024-08-07

We thank all reviewers for their detailed, insightful, and overwhelmingly positive reviews.
We are encouraged to see that the reviewers consider the problem we investigate under-studied and important (PhkB, i5fN, pxq3, ueFW), our benchmark impactful (PhkB, i5fN, Ky1F, pxq3, ueFW), our empirical evaluation and its analysis exhaustive and interesting (PhkB, i5fN, Ky1F, pxq3), and our exposition clear and well written (PhkB, pxq3). Below, we address remaining shared questions, before going into more detail in reviewer-specific responses.

Since the original submission of this work, we have fully dockerized our evaluation, improved the parsing of test results to cover additional edge cases, and conducted experiments on more models (GPT4o-mini, Claude 3.5 Sonnet and Mistral Large 2, all released after the submission), methods (Aider), and ablation settings (sampling temperature, agent interaction rounds, multiple runs). Please see the attached PDF for these updated and extended results, referred to in some of our responses.

**Statistical significance of results (Ky1F, pxq3)**
To confirm the statistical significance of our results, we have conducted a dependent t-test for paired samples to compute the statistical significance of SWE-Agent+ having higher performance than LIBRO. We find that SWE-Agent+ has higher performance (Fail-to-Pass rate) with a p-value of $7 \times 10^{-6}$, indicating strong statistical significance of this main result. We further want to highlight that most of the considered approaches use greedy decoding and are thus deterministic up to floating point errors. Finally, we conduct additional experiments with GPT4o-mini using sampling at a range of temperatures and consistently find small 95% confidence intervals of around $\pm$0.2% over 5 runs of ZeroShotPlus, demonstrating only small sensitivity to inference randomness even when not using greedy decoding. Further details are provided in Table 3 of the attached PDF. We believe these results further strengthen our conclusions and are happy to include them in the main paper.

**Relevance of Python results and generalization to other languages (Ky1F, pxq3, ueFW)**
First, we want to highlight that Python is an extremely popular language, often ranked #1 by a wide margin in corresponding surveys [1][2]. Thus, we believe that our results on Python are already highly relevant. Further, our approach to designing a testing benchmark can be applied to any programming language where Github repositories with suitable issues and pull requests can be sourced. However, setting up the corresponding execution environments with historically correct dependency versions and test results parsing requires some manual effort. Given the already substantial cost of evaluating the whole benchmark of 1700 instances, we decide to leave this to future work targeting specific languages or issue types.

**What is the effect of possible data contamination on SWT-Bench and how can it be addressed? (Ky1F, ueFW)**
As SWT-Bench is based on historic GitHub issues, they could be included in the pre-training data of the LLMs underlying the methods we investigate. To assess this effect, we conducted an experiment comparing the performance (of ZeroShotPlus) on all issues created after the training data cutoff of GPT4 (April 2023) to a random subset of instances created before of the same size and include the results in the attached PDF in Table 4. We observe that for the issues created before the cutoff, only one more sample is solved compared to those created after, leading to no statistically significant difference in performance (p-value of 37% using Students t-test). This agrees well with the findings on SWE-Bench. We are happy to include a corresponding discussion and these results in the revised version of our paper. We further note that all methods we investigate should benefit from memorization to a similar extent and hence contamination should not affect their ranking and our conclusion that Code Agents perform surprisingly well.

One approach to address this contamination issue is to create a rolling version of SWT-Bench, based only on the most recent issues. However, this comes at the cost of direct comparability of results and increased cost for reproducing results for all baselines on a changing evaluation set.

**Conclusion**
We hope to have been able to address the reviewers’ questions and look forward to the discussion period.

**References**
[1] TIOBE Index for August 2024, [https://www.tiobe.com/tiobe-index/](https://web.archive.org/web/20240807025036/https://www.tiobe.com/tiobe-index/)
[2] PYPL PopularitY of Programming Language, [https://pypl.github.io/PYPL.html](https://web.archive.org/web/20240806100838/https://pypl.github.io/PYPL.html)

---

### Comment · Area_Chair_7iM4 · 2024-08-08
**Author-Reviewer discussion (Aug 7 - Aug 13)**

Dear Submission15335 reviewers,

We appreciate your reviews as we move into the Author-Reviewer discussions phase (Aug 7 - Aug 13).
Please read all reviews and author responses carefully.
Please address any remaining questions or concerns with the authors and respond to their rebuttal as needed. Authors need time to respond to your messages, so please post your responses as soon as possible, so there is time for back-and-forth discussion with the authors. At a minimum, please acknowledge that you have read the author rebuttal. Based on the material provided, please adjust your scores and reviews as necessary.

Dear Submission15335 authors,

We appreciate your rebuttal. Please continue the dialogue with the reviewers during this discussion phase.

This message and thread are visible to both reviewers and authors. If you need to respond privately, adjust the "Readers" setting accordingly.

---

### Comment · Area_Chair_7iM4 · 2024-08-08
**Questions for authors**

Dear authors,

Thank you for your rebuttal.
I have couple questions about your submission that may clarify certain aspects of your paper and facilitate the discussion with reviewers.

Page 3:
"Golden Tests" -> are these tests that were added in the patch under consideration?

Page 4:
"If none of the lines modified by the golden patch X are executed by any test, i.e., |X ∗r | + |X ∗166 a | = 0, we exclude this instance from our coverage analysis (43.1% of the cases)." - can you elaborate on this? How does the patch fix an issue if none of the tests execute any lines in it? Is the patch deleting erroneous code? 43% seems like a large percentage of cases for such occurrences.

Page 7, table 2:
What is "valid patch" in section 5.2? Is it a test case patch that is valid? You are not talking about a code patch that fixes the issue, right? How is the validity checked? Maybe consider renaming the "valid patch", since it might be misleading for readers when test change is named "patch" the same way as code fix change is named "patch".

What is fail-to-any tests (F→×)? Why are they considered to be good? Can you elaborate? Isn’t there a risk that LLM generates a meaningless test that fails both on initial codebase and on the patched codebase and therefore does not contribute a valid test?

Thank you

---

> ### Author Response · Authors · 2024-08-09
> **Response to the AC**
>
> We thank the AC for taking such an active role in chairing the discussion and are happy to answer their questions below:
>
> **What are golden tests?**
> Exactly as the AC suggests, these are the tests that were added in the pull request that “resolved” the GitHub issue underlying the SWT-Bench instance. At least one of these tests fails on the original state of the codebase, before the golden code patch is applied, and passes after. They are thus always a valid (gold standard) solution for the SWT-Bench task. We will make this more clear in Line 32 where they are first introduced.
>
> **Can you discuss what can cause none of the lines modified by the golden code patch to be covered by the test suite?**
> The main reason for this is that Python’s _trace_ module by default is unable to measure coverage in subprocesses. If (part of) a test suite is executed using such independent processes, coverage can not be measured accurately and no execution of the patched code is reported. Meanwhile, threading based concurrency is handled correctly. We would like to highlight that this issue comes from the task instance and Python’s _trace_ module, but not the test generation method. The measured coverage is thus still comparable across different test generation methods and computed over the remaining 57% of SWT-Bench instances.
>
> **Can you clarify what a _valid_ patch is in Section 5.2?**
> A valid patch is one that can be successfully/without error applied to a codebase, i.e. that has the correct format. As generated tests are added to the code base in the form of a patch, we reused the term “valid patch”, currently defined implicitly in Line 170. We will follow the AC’s suggestion and rename them to “well-formed tests” and introduce this terminology explicitly to improve clarity. As many models struggle to produce valid/well-formed (i.e. applicable without error) test patches in the default unidiff standard, the rate of valid test patches is a relevant metric in particular when comparing our novel (ZeroShotPlus) with the standard unidiff patch format (ZeroShot).
>
> **What is a fail-to-any (F -> X) test and why is it “good” to measure this?**
> A fail-to-any test is one that fails on the original codebase and either passes or fails on the codebase with the golden patch applied. While some F->X tests might be undesirable as the AC suggests, we consider the rate of F->X tests as a useful metric for assessing a test agent’s overall capabilities for two main reasons. First, creating an F->X test is a prerequisite for generating a fail-to-pass (F->P) test, which is ultimately the desired outcome. Therefore, the rate of F->X serves as an intermediate metric for our final F->P metric. Second, given that the agent has access to the original version of the codebase and the issue report, it can leverage these two inputs to self-verify if a test fails on the codebase before proposing the test. The rate of F->X tests reflects the agent’s ability to perform this self-verification. We will add this discussion in the corresponding section.
>
> **Conclusion**
> We hope to have been able to address the AC’s questions and remain happy to answer any follow-up they might have.

---

> > ### Comment · Area_Chair_7iM4 · 2024-08-10
> > **Thank you**
> >
> > Thank you for your responses.

---

### Comment · Area_Chair_7iM4 · 2024-08-12
**Reviewer responses to authors (Aug 7 - Aug 13)**

Dear Submission15335 reviewers,

We appreciate your reviews. We have only two days left for the author-reviewer discussions. Please read all reviews and author responses. It is helpful if you respond to authors' rebuttal and address any remaining questions or concerns with the authors. At a minimum, please acknowledge that you have read the author rebuttal. If your evaluation of the paper has changed, please adjust your scores and reviews as necessary.

Thank you.

---

### Decision · Program_Chairs · 2024-09-25

**Decision:**

Accept (poster)

**Comment:**

This paper has a very wide spread of scores from 3 (reject) to 8 (strong accept). Unfortunately, two reviewers who suggest accept and strong accept did not participate in discussions. The issues raised by reviewer who suggested reject do not seem significant to me apart from data contamination (see below). After reading the paper, reviews, rebuttal, and substantial discussions between authors and reviewers, I believe that the paper should be accepted.

Summary of Strengths:

- Novel Benchmark: The paper introduces a new benchmark (SWT-Bench) for evaluating automated test generation, particularly using Code Agents and LLMs. This is recognized as filling a gap in current research.
- Practical Relevance: The benchmark uses real-world data from popular Python repositories, making it relevant and practical for testing the capabilities of LLMs in generating tests.
- Detailed Analysis: The evaluation is comprehensive, with multiple metrics (e.g., fail-to-pass rate, code coverage) and methods compared, showing the potential impact of automated test generation on software quality and developer productivity.

Weaknesses identified:

- Limited Statistical Analysis: The statistical analysis is seen as insufficient, particularly due to the computational constraints that prevented a full analysis. This limitation affects the confidence in the reported results.
- Focus on Python: The benchmark focuses solely on Python, which raises concerns about the generalizability of the results to other programming languages.
- Potential Data Contamination: The benchmark might include data from GitHub that was part of the pre-training corpus for LLMs, potentially contaminating the evaluation results.
- Limited Exploration of Hyperparameters: The paper does not fully explore the impact of different hyperparameters on the performance of Code Agents, missing potential optimization opportunities.

Authors' Response to Weaknesses:

- Statistical Analysis: Additional experiments were conducted to demonstrate the statistical significance of their results, which were reported in the rebuttal.
- Python Focus: The authors explained that while the current work focuses on Python, their approach is applicable to other programming languages, and expanding to other languages is left for future work.
- Data Contamination: The authors provided additional experiments to assess the effect of data contamination and found no significant impact on their results. However these experiments are based on small data sizes. This remains a potential issue.
- Hyperparameter Exploration: An ablation study on key hyperparameters was conducted and shared in the rebuttal, providing insights into optimizing the agents.

In terms of weaknesses, I think that only data contamination has not been fully addressed and is possibly concerning with respect to paper's results.